# Thermal phenotypic plasticity of pre- and post-copulatory male harm buffers sexual conflict in wild *Drosophila melanogaster*

**Claudia Londoño-Nieto[1]\*, Roberto García-Roa[1,2], Clara Garcia-Co[1], Paula González[1], Pau Carazo[1]**

[1]Ethology Lab, Cavanilles Institute of Biodiversity and Evolutionary Biology, University of Valencia, Valencia, Spain; [2]Department of Biology, Lund University, Lund, Sweden

**\*For correspondence:**
claudia.londonon@gmail.com

**Competing interest:** The authors declare that no competing interests exist.

**Abstract** Strong sexual selection frequently leads to sexual conflict and ensuing male harm, whereby males increase their reproductive success at the expense of harming females. Male harm is a widespread evolutionary phenomenon with a strong bearing on population viability. Thus, understanding how it unfolds in the wild is a current priority. Here, we sampled a wild *Drosophila melanogaster* population and studied male harm across the normal range of temperatures under which it reproduces optimally in nature by comparing female lifetime reproductive success and underlying male harm mechanisms under monogamy (i.e. low male competition/harm) vs. polyandry (i.e. high male competition/harm). While females had equal lifetime reproductive success across temperatures under monogamy, polyandry resulted in a maximum decrease of female fitness at 24°C (35%), reducing its impact at both 20°C (22%), and 28°C (10%). Furthermore, female fitness components and pre- (i.e. harassment) and post-copulatory (i.e. ejaculate toxicity) mechanisms of male harm were asymmetrically affected by temperature. At 20°C, male harassment of females was reduced, and polyandry accelerated female actuarial aging. In contrast, the effect of mating on female receptivity (a component of ejaculate toxicity) was affected at 28°C, where the mating costs for females decreased and polyandry mostly resulted in accelerated reproductive aging. We thus show that, across a natural thermal range, sexual conflict processes and their effects on female fitness components are plastic and complex. As a result, the net effect of male harm on overall population viability is likely to be lower than previously surmised. We discuss how such plasticity may affect selection, adaptation and, ultimately, evolutionary rescue under a warming climate.

## Editor's evaluation

This study has important implications for the impact of sexual conflict on population viability under different temperatures. The authors provide compelling evidence that male harm to females in sexual conflict can be reduced as a function of temperature within the optimal reproductive range of a species. The results have implications for the likelihood of the evolutionary rescue of species facing the climate crisis.

## Introduction

Females and males share the common goal of siring offspring. This central tenet of sexual reproduction enforces a certain degree of cooperation between the sexes. However, anisogamy frequently leads to distinct sex roles and thus general asymmetries in the reproductive evolutionary interests of females and males, which can in turn result in diverging intensity and form of sexual selection across the sexes (*Arnqvist and Rowe, 2005*; *Chapman et al., 2003a*; *Janicke et al., 2016*; *Winkler et al.,*

*2021*). This phenomenon, termed sexual conflict, favors traits in one sex that might be costly for the other (*Parker, 1979*), and can thus lead to antagonistic female-male coevolution (*Arnqvist and Rowe, 2005*). Sexually antagonistic co-evolution has received much attention and is recognized as a fundamental process in evolution due to its role in shaping male and female adaptations (*Bonduriansky et al., 2008*), in contributing to drive reproductive isolation and speciation (*Arnqvist et al., 2000*; *Bonduriansky, 2011*; *Bonduriansky and Chenoweth, 2009*; *Gavrilets, 2014*), and as a major determinant of population demography (*Kokko and Brooks, 2003*; *Bonduriansky and Chenoweth, 2009*; *Berger et al., 2016*). Specifically, sexual conflict has been shown to have profound consequences for female fitness and population growth when it favors male reproductive traits that increase male intra-sexual competitive ability at the expense of harming females (i.e. male harm, *Crudgington and Siva-Jothy, 2000*; *Gómez-Llano et al., 2023*; *Wigby and Chapman, 2004*).

Harmful male adaptations are widespread and incredibly diverse and sophisticated across the tree of life (*Arnqvist and Rowe, 2005*). For example, male harassment of females during pre-copulatory competition for mating has been documented in myriad vertebrate and invertebrate species (*Gómez-Llano et al., 2023*), driving antagonistic female-male co-evolution in a host of behavioral and morphological traits (*Arnqvist and Rowe, 2005*). Male harm adaptations in the context of post-copulatory competition are similarly widespread in invertebrates, featuring (amongst others) toxic ejaculates (*Wigby and Chapman, 2005*), love darts (*Koene and Schulenburg, 2005*), and a range of male adaptations for traumatic insemination that range from genital ablation to spiny penises (*Crudgington and Siva-Jothy, 2000*; *Lange et al., 2013*). Importantly, beyond driving female and male phenotypes and associated diversification processes, male harm generally leads to a 'reproductive tragedy of the commons' that can substantially impact population demography by depressing net female productivity (*Arnqvist and Tuda, 2010*; *Berger et al., 2016*; *Holland and Rice, 1999*; *Rankin et al., 2011*), and even facilitate population extinction (*Le Galliard et al., 2005*). Understanding what factors underlie male harm evolution, its diversity in form, strength, and outcomes, is thus a main concern in evolutionary biology.

Despite a growing number of studies in the field of sexual conflict, most have been conducted under uniform laboratory conditions, frequently in populations adapted to stable environments for hundreds of generations (*Chapman et al., 2003b*; *Hopkins et al., 2020*; *Wigby and Chapman, 2004*). In contrast, recent research has highlighted the role of ecology in shaping the evolution of traits under sexual conflict (*Arbuthnott et al., 2014*; *García-Roa et al., 2019*; *MacPherson et al., 2018*; *Perry and Rowe, 2018*; *Yun et al., 2017*), including habitat complexity (*Malek and Long, 2019*; *Miller and Svensson, 2014*; *Myhre et al., 2013*), nutritional status (*Fricke et al., 2010*), or sex ratio and population density (*Chapman et al., 2003a*). For example, *Gomez-Llano et al., 2018* recently showed that conspecific densities and the presence of heterospecifics modify the intensity and outcome of sexual conflict in the banded demoiselle (*Calopteryx splendens*), and the spatial complexity of the environment in which mate competition occurs mediates how sexual conflict operates in fruit flies (*Yun et al., 2018*). The incorporation of more realistic ecological scenarios in the study of sexual conflict is thus a key avenue to disentangle the evolution of male harm, and its consequences for population viability (*Cornwallis and Uller, 2010*; *Fricke et al., 2009*; *Plesnar-Bielak and Łukasiewicz, 2021*).

Temperature is recognized as a crucial abiotic ecological factor due to its impact on life history traits and physiological and behavioral responses (*De Lisle et al., 2018*; *Kim et al., 2020*; *Miler et al., 2020*; *Monteiro et al., 2017*). Furthermore, temperature varies in nature widely within and across spatiotemporal scales (e.g. daily, inter-seasonal, and intra-seasonal variation). Consequently, it may have short, medium, and long-term effects on organism phenotypes that can impact many different aspects of its reproductive behavior (e.g. sex-specific potential reproductive rates, operational sex ratios, density, etc.; *García-Roa et al., 2020*). In fact, a recent meta-analysis suggests that temperature may have a sizeable effect on sexual selection processes even when fluctuations occur well within the normal range of temperature variation for the studied species (*García-Roa et al., 2020*). This latter finding is particularly relevant given that we know almost nothing about how average temperature fluctuations, such as those experienced by wild populations during their reproductive season, affect male harm and sexual selection at large.

Our aim was to contribute to fill this gap in knowledge by studying how male harm responds to temperature variation that mimics average fluctuations that are normal during the reproductive season, using *Drosophila melanogaster* flies sampled from a wild population. *D. melanogaster* is an ideal

subject for this study because it exhibits high levels of male-male competition, has well-characterized pre- and post-copulatory male harm mechanisms, and is perhaps the main model species in the study of sexual conflict (*Arbuthnott et al., 2014*; *Chapman et al., 2003a*; *MacPherson et al., 2018*; *Malek and Long, 2019*; *Wigby and Chapman, 2005*). During male-male pre-copulatory competition over access to mating, males harm females via intense harassment that causes physical injuries, interferes with female behaviors such as egg-laying or feeding, and results in energetically costly resistance to males (*Bretman et al., 2019*; *Partridge and Fowler, 1990*; *Teseo et al., 2016*). In the context of sperm competition over fertilizations, males can also harm females via toxic ejaculates, whereby certain male seminal fluid proteins manipulate female re-mating and egg-laying rates to the male's advantage, but at a cost to females in terms of lifespan and lifetime reproductive success (*Chapman et al., 2003b*; *Sirot et al., 2009*; *Wigby and Chapman, 2005*). These proteins are secreted by male accessory glands, have been well characterized, and are strategically allocated by males in response to variation in the socio-sexual context (*Hopkins et al., 2019*; *Sirot et al., 2011*). Additionally, despite the fact that almost all work on male harm has, to date, been conducted in laboratory strains adapted to stable temperature conditions, *D. melanogaster* reproduces in the wild under a wide range of temperatures (*Dukas, 2020*; *Kapun et al., 2018*).

Briefly, we collected flies from a continental wild population in Requena (Spain) that experiences significant fluctuations in temperature even during the mildest months when it is reproductively active (e.g. July: average: 24.9°C, average min: 19.8°C, average max: 30.1°C, *Fick and Hijmans, 2017*). After acclimation of the resulting population to laboratory conditions under a fluctuating temperature regime mimicking natural conditions, we conducted five different experiments to gauge how

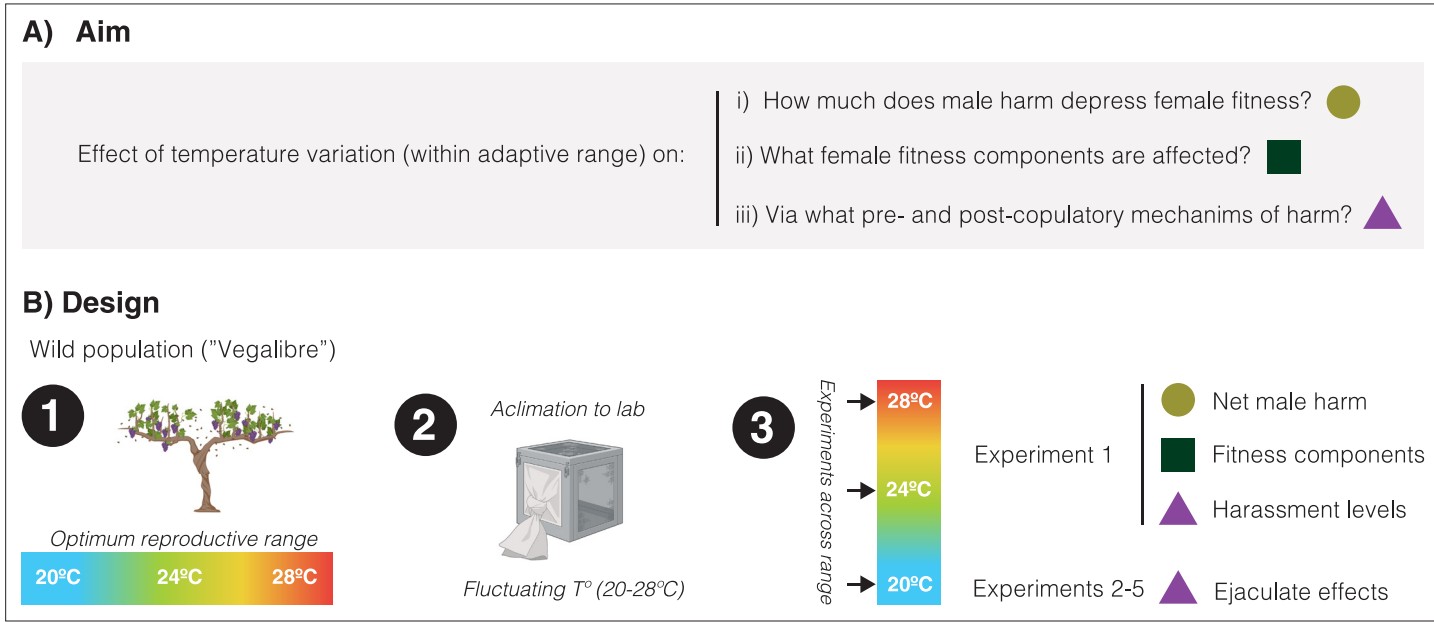

**Figure 1.** Schematic overview of the study. (**A**) Our aim was to study how temperature variation, across a range at which reproduction is optimum in the wild, may affect: the net decrease in female fitness resulting from male harm, what female fitness components are mainly affected by male harm, and pre-copulatory (i.e. sexual harassment) and post-copulatory (i.e. ejaculate effects on female receptivity, short-term fecundity, and survival) mechanism of harm. (**B**) General design of the study: (1) We sampled a wild population of *Drosophila melanogaster* flies that reproduce optimally between 20°C and 28°C, (2) We setup a population in the lab and left it to acclimate for a few generations under a programmed fluctuating temperature regime that mimics wild conditions in late spring-early summer (20–28°C range with mean at 24°C), (3) We run a series of five experiments (each repeated at 20°C, 24°C, and 28°C) to study temperature effects on net male harm, female fitness components and male pre- and post-copulatory mechanisms of harm.

The online version of this article includes the following figure supplement(s) for figure 1:

**Figure supplement 1.** Fitness and behavioural assay design (Experiment 1).

**Figure supplement 2.** Receptivity assay design (Short treatment duration – 48 hr, experiment 2).

**Figure supplement 3.** Receptivity assay design (Long treatment duration – 13 days, experiment 3).

**Figure supplement 4.** Fecundity and survival assay design (Short treatment duration – 48 hr, experiment 4).

**Figure supplement 5.** Fecundity and survival assay design (Long treatment duration – 13 days, experiment 5).

temperature variation within a normal range (i.e. 20°C, 24°C, and 28°C) affects: (a) the overall impact of male-male competition on female lifetime reproductive success (i.e. male harm), (b) how the net effects of harm are accomplished in terms of different female fitness components (i.e. reproductive rate, actuarial aging, and reproductive aging), and (c) underlying male pre-copulatory (i.e. harassment) and post-copulatory (i.e. ejaculate toxicity) harm mechanisms (*Figure 1*).

## Materials and methods

### Field collection

In October 2018, we used banana traps to sample *Drosophila melanogaster* flies from three wineries in Requena (Spain): 'Hispano-Suizas' (39.466128,–1.149642), 'Pago de Tharsys' (39.497834,–1.122781), and 'Dominio de la Vega' (39.515079,–1.143757). Traps were setup within the wineries, but in premises that were open to the exterior and to ample surrounding vineyards and/or in the vineyards themselves. After collection, we anesthetized flies using mild $CO_2$ exposure. We then separated and individually distributed field-collected females in vials with standard food, left them to lay eggs for a period of 48 hr, and then incubated their eggs at 24°C, 60% humidity, and a 12:12 dark-light cycle for 14 days to allow adult flies to emerge. We inspected the genital arch of F1 males of each of these female isolines to distinguish *D. melanogaster* isofemale lines from *D. simulans*. We then collected three males and three females from each *D. melanogaster* isofemlae line (total of 276 flies from 46 isofemale lines) and released them into a population cage with a surplus of food medium supplemented with live yeast, setting up the 'Vegalibre (VG)' population. In November 2019 and October 2020, to maintain natural variation, we re-sampled the wineries and added 348 and 756 flies from 58 and 126 isofemale lines, respectively, following the same procedure (27 isofemale lines from 'Pago de Tharsys' and 31 isofemale lines from 'Dominio de la Vega' in November 2019 and 33 isofemale lines from 'Pago de Tharsys,' 50 isofemale lines from 'Dominio de la Vega' and 43 isofemale lines from 'Hipano-Suizas' in October 2020).

### Stock maintenance and acclimation

We carried out all experiments between March 2020 and April 2021, using individuals from the VG field population kept in the laboratory with overlapping generations at an average temperature of 24°C with daily pre-programmed fluctuations (±4°C) mimicking natural daily temperature conditions during the reproductively active season, at ~60% humidity and on a 12:12 hr light:dark cycle (Pol Eko ST 1200 incubator). The lowest temperature was set up 1 hr after sunrise and the highest 1 hr after midday. It is important to note that our stock population of flies was kept under a programmed fluctuating temperature regime that mimics their average circadian rhythm in the field, but temperature fluctuations in nature will be inherently subject to minor stochastic variations whose effects we controlled for (and thus did not capture) in this experiment. We used maize-malt medium (7 g of agar, 72 g of malt, 72 g of maize, 9 g of soya, 15 g of live yeast, 20 g of molasses, and 33 ml of Nipagin mix –3 g of methyl 4-hydroxy-benzoate and 28 ml of ethanol 90%– per 1000 ml of water) as a food source throughout maintenance and experiments. To collect experimental flies, we introduced yeasted grape juice agar plates into stock populations to induce female oviposition. We then collected eggs and placed them in bottles containing ~75 ml of medium to be incubated at 24 ± 4°C at a mean density of 223 ± 14.3 (95% CI) (*Clancy and Kennington, 2001*). We collected virgin flies within 6 hr of eclosion, under ice anesthesia, and then sexed and kept them in vials with food until their use (~3 days later), at 24 ± 4°C (see below for more details).

### Net impact of male harm on female fitness and underlying behavioural mechanisms (experiment 1)

#### Fitness assay

To study whether male harm is affected by temperature, we established a factorial design to measure survival and lifetime reproduction success (LRS) of female flies under monogamy (i.e. one male and one female per vial) vs. polyandry (i.e. three males and one female per vial), across three stable temperature treatments typical of this population during their reproductively active period in the wild: 20°C, 24°C, and 28 °C. Comparison of female fitness at monogamy vs. polyandry is standard procedure to gauge male harm in *Drosophila* and other organisms (see *Yun et al., 2021* for a review

and *Gómez-Llano et al., 2023* for a recent meta-analysis). While the sex ratio in this species is typically 1:1, the operational sex ratio is male-biased and frequently reaches a 3:1 (or higher) male-bias in mating patches in the wild (*Dukas, 2020*). Thus, the 1:1 vs. 3:1 sex ratios used in this study represent biologically relevant scenarios and have actually become standard in *Drosophila* studies measuring male harm to females (*Yun et al., 2021*; *Gómez-Llano et al., 2023*).

We first collected virgin flies into same-sex vials of 15 individuals and then randomly allocated them to either of the three temperature treatments 48 hr before starting the experiment, at which temperatures they remained until its end. To estimate LRS, we transferred flies to fresh vials twice a week using mild $CO_2$ anesthesia. We incubated the vials containing female eggs at 24 ± 4°C for 15–20 days (~15 days for vials coming from 28°C, ~17 days for 24°C and ~20 days for 20°C) to allow F1 offspring emergence, after which we froze them at –21°C for later counting. The differences in incubation time are due to differences in developmental time caused by temperature differences during the first 3–4 days of each vial (i.e. the time eggs remained at their respective temperature treatments before flipping females to new fresh vials and incubation at 24 ± 4°C). We discarded and replaced males with young (2–4 days old) virgin males (receiving the same treatment as described above for original males) three weeks after starting the experiment (at the same time for all treatments). In addition, we kept a stock of replacement males maintained at each of the three temperatures to replace dead male flies if needed. We kept focal female flies under these conditions for six weeks, after which we discarded males and followed females until they died for survival analysis (see *Figure 1—figure supplement 1* for an overview of the experimental design).

We started the experiment with 468 females (78 per temperature and mating treatment) and 936 males (234 per temperature in polyandry and 78 per temperature in monogamy). Due to discarded (e.g. accidentally damaged during handling) and escaped flies, final (female) sample sizes were: (a) at 20°C, $n_{polyandry} = 74$ and $n_{monogamy} = 76$, (b) at 24°C: $n_{polyandry} = 72$ and $n_{monogamy} = 77$, and (c) at 28°C: $n_{polyandry} = 70$ and $n_{monogamy} = 75$. We estimated the overall degree of male harm by calculating relative harm (H) following *Yun et al., 2021*:

$$H = \frac{W_{monogamy} - W_{polyandry}}{W_{monogamy}}$$

where W corresponds to female fitness. Thus, H provides an estimate of the relative decrease in female fitness due to male harm.

Using the data collected above, we partitioned overall LRS effects into effects on early reproductive rate (i.e. offspring produced during the first two weeks of age), actuarial aging (i.e. lifespan), and reproductive aging (i.e. offspring produced over weeks 1–2 vs. 3–4). We used weeks 3–4 as an estimate of late reproductive rate because mortality was already evident at this point (i.e. reflecting aging) and then was very high from week 5 onwards (*Figure 4—figure supplement 1*; thus preventing accurate estimation of reproductive success).

Finally, we also calculated rate-sensitive fitness estimates for each individual female and treatment population. Rate-sensitive fitness estimates take into account when offspring are produced, not just how many offspring are produced, and thus allow estimating fitness subject to the population growth rate (*Edward et al., 2011*). It is important to understand how differences in the number and timing of offspring production translate into fitness under different demographic scenarios. For example, early reproduction is particularly favored in increasing populations whereas late reproduction gains in importance in decreasing populations. Thus, while LRS is most suited to estimate individual fitness in stable populations, rate-sensitive estimates are preferred when r ≠ 0 (*Brommer et al., 2002*). We calculated both individual ($\omega_{ind}$) and population ($\omega_{pop}$) rate-sensitive fitness for the following intrinsic rates of population growth: r=–0.1, r=–0.05, r=0, r=0.05, and r=0.1 (see *Edward et al., 2011* for a detailed account). We then used $\omega_{pop}$ values to calculate the relative cost ($C_r$) of polyandry for each temperature treatment across different values of r as:

$$Cr = \frac{\omega_{pop\ (polyandry)}}{\omega_{pop\ (monogamy)}}$$

## Behavioral measures

Immediately after the fitness experiment started, we conducted behavioral observations on the first day of the experiment across all treatments (*Figure 1—figure supplement 1*). Our aim was to investigate the behavioral mechanisms that might underlie the potential fitness effects evaluated above. Due to logistic limitations, we conducted behavioral observations in the same temperature control room, so we had to conduct trials at 20°C, 24°C, and 28°C over three consecutive days (with both monogamy and polyandry treatments evaluated at the same time for each temperature), in a randomized order (i.e. 20°C, 28°C, and 24°C). Note that we collected virgin flies over three consecutive days to ensure all flies were 5 days-old at the start of the experiment. We recorded the following behaviors: (a) courtship intensity (number of courting males per female per hour), (b) male-male aggression rate (i.e. number of aggressions per hour), and (c) female rejection (i.e. number of rejections per hour; see *Bastock and Manning, 1955*; *Connolly and Cook, 1973* for behavioral descriptions). We also recorded the number of total matings during the observation period.

Observations started at lights-on (10 a.m.) and lasted for 8 hr, during which time we continuously recorded reproductive behaviors using scan sampling of vials. Each complete scan lasted approximately 8 min, so that we always conducted one complete scan every 10 min to ensure the recording of all matings (see below). Scans consisted in observing all vials in succession for ca. 3 s each and recording all occurrences of the behaviors listed above (i.e. all-occurrences recording of target behaviors combined with scan sampling). We interspersed behavioral scans with very quick (<1 min) mating scans where we rapidly swept all vials for copulas at the beginning, in the middle, and at the end of each complete scan. This strategy ensured that we recorded all successful matings (>10 min), which typically last between 15 and 25 min in our population of *D. melanogaster*. We obtained a total of 49 scans per vial. Behavioral observations were conducted only once, on day 1 of the fitness experiment, as prior experiments have shown that courtship, aggressive and female rejection behaviors in *D. melanogaster* are sufficiently stable over time so that our behavioral indexes are representative of long-term treatment differences (e.g. *Carazo et al., 2015*; *Carazo et al., 2014*). In contrast to courtship, aggression, and rejection indexes, note that total mating frequency over the first day cannot be taken as a reliable measure of mating rate (*Wolfner, 1997*). Thus, our rationale in recording this variable was just to ensure that early mating ensued normally across treatments (which was the case, see *Figure 5—figure supplement 2*), but we did not include this variable in our statistical analyses.

## Mating effect on female reproduction and survival (experiments 2 to 5)

To examine post-mating mechanisms that might underlie the fitness effects observed in our first experiment, we conducted four additional experiments to test whether temperature modulates the well-documented effects that mating with a male has on female receptivity, short-term fecundity, and survival. In *D. melanogaster*, males manipulate female reproduction via their ejaculate, which increases male fitness but frequently decreases female lifespan and lifetime reproductive success (*Chapman et al., 1995*). Briefly, males transfer seminal fluid proteins (SFPs) produced by their accessory glands that stimulate female short-term fecundity, decrease female receptivity, and ensure sperm storage, thus generally promoting male success in sperm competition (*Chapman et al., 1995*; *Wigby and Chapman, 2005*). In addition, prior studies have shown that males are able to tailor investment into SFPs according to perceived sperm competition risk (SCR) and intensity (*Hopkins et al., 2019*). Thus, we set-up a factorial design where we manipulated the temperature (i.e. 20°C, 24°C, and 28°C) and perceived SCR levels (i.e. males kept alone vs. with 7 more males in a vial) at which adult focal males were kept prior to mating.

Then, we measured how the reception of a treated male's ejaculate after a single mating in a common garden environment (i.e. all matings at 24 °C) affected female fecundity, survival, and reproduction, following standard assays to gauge male ejaculate effects on females in *D. melanogaster* (e.g. *Chapman et al., 1995*; *Perry et al., 2013*; *Wigby and Chapman, 2005*; *Wigby et al., 2009*). We conducted separate experiments implementing two different temperature treatment durations (i.e. 48 hr and 13 days), to include two potential different scenarios. Our 48 hr treatment aimed to mimic short-term temperature effects on adult males whereas our 13 day treatment aimed to mimic longer-term effects on adult males that span a complete spermatogenesis cycle. The period from the synthesis of deoxyribonucleic acid in the spermatocyte to successful insemination is approximately 10 days long in *D. melanogaster* (*Chandley and Bateman, 1962*), so we treated males for 13 days after

sperm depleting them (see below) to ensure that males experienced treatment temperatures across the whole spermatogenesis cycle.

## Receptivity assays (experiments 2 and 3)

We first collected focal males as virgins (i.e. within 6 hr of eclosion) under ice anesthesia and randomly placed them either individually (low SCR) or in a same-sex group of eight (high SCR) in plastic vials with food. Next, we randomly divided them into three groups that we allocated to the different stable temperature treatments for either 48 hr (i.e. short treatment duration, experiment 2, *Figure 1—figure supplement 2*) or 13 days (i.e. long treatment duration, experiment 3, *Figure 1—figure supplement 3*) immediately before the beginning of each experiment. For experiment 3, we depleted the sperm and seminal fluid of focal males before allocating them to different temperature treatments by housing them with four standard virgin females for 24 hr, given that three successive matings are enough to deplete the accessory glands of male *D. melanogaster* (*Linklater et al., 2007*; *Macartney et al., 2021*).

We collected all females and competitor males (i.e. standard males without any previous treatment) used in receptivity assays as virgins and held them in same-sex groups of 15–20 flies at $24 \pm 4°C$. Experiments started by exposing all virgin females to single focal males for 2.5 hr at 24°C. After a successful copulation, we separated the mated females from the focal males and isolated them until the remating trial. We discarded unmated females and focal males. 72 hr after this first mating with the focal treated male, we individually exposed females to single virgin competitor males for 12 hr. After each trial, we transferred unmated females into a new vial with food, until the next remating trial on the next day (*Figure 1—figure supplements 2 and 3*). We repeated remating trials for three consecutive days, which allowed us to calculate the cumulative percentage of remated females (and associated re-mating latencies; see below) for each of the three days of each experiment. Due to a large number of vials/flies involved, we conducted the experiments in two blocks each: with n=390 females per batch in experiment 2 (n=436 rematings) and n=420 females per batch in experiment 3 (n=676 rematings). We also recorded mating duration for the first mating (i.e. with the focal treated male), the remating latency (i.e. the time lapse between males being introduced into the female-containing vial and copulation), and mating duration for re-matings. Females and focal and competitor males were 4 days old for experiment 2. In experiment 3, females and competitor males were 4 days old, while focal males were 18 days old.

## Fecundity and survival assays (experiments 4 and 5)

To study the effects of a single mating on female short-term fecundity and long-term survival, we performed two experiments (experiments 4 and 5, *Figure 1—figure supplements 4 and 5*, respectively) where we compared female fecundity and F1 egg-to-adult viability of females mated with male flies subject to the same factorial design imposed in receptivity experiments (here experiment 4 had a treatment duration of 48 hr while experiment 5 had a treatment duration of 13 days). We collected and treated all focal males as in the receptivity assays described above, and then proceeded to mate virgin females in single pairs with focal males for 2.5 hr at 24°C. After copulation, we separated mated females from focal males and kept them individually in single vials. We discarded unmated females and focal males. We then transferred females to fresh vials every 24 hr for 4 days, and then every 3 days twice. Finally, we followed females until they died by combining them into same-treatment vials of 10 females that were flipped once a week. We removed dead flies at each flip and scored deaths on a daily basis. We counted eggs laid during the first 3 days and incubated vials from days 1, 2, 3, 4, 5, and 8 until adults emerged to count progeny and determine egg-to-adult viability (*Figure 1—figure supplements 4 and 5*). Sample sizes were 545 females for experiment 4, and 480 females for experiment 5. Females and focal or competitor males were 4 days old for experiment 4. In experiment 5, females and competitor males were 4 days old, while focal males were 18 days old.

## Statistical analyses

We performed all statistical analyses using R statistical software (version 3.5.2). In all cases, we assessed fit and validated models by visual inspections of diagnostic plots on raw and residual data (*Zuur et al., 2010*). In all models, we used ANOVA type III test 'F' to compute *p*-values corrected by the Benjamini-Hochberg (BH) method to control the inflation of the type I error-rate due to multiple

testing. We fitted all models with the temperature effect as a covariate, given that it is a continuous variable. In all cases where we detected a significant interaction between main effects, we ran models separately for each temperature (or treatment duration for experiments 2–5) to explore the nature of such interactions. As a complementary analysis, in these cases, we re-fitted the original model with temperature as a factor, which allowed us to run a post hoc Tukey's test as an additional way to explore interactions while controlling for inflation of experiment-wise type 1 error rate. We provide the latter in the SM, but we note that both approaches always yielded qualitative identical results.

### Experiment 1

To examine temperature effects on male harm, we evaluated the interaction between mating system and temperature on female fitness (LRS), early reproductive rate, reproductive aging, actuarial aging, and male and female reproductive behaviors (courtship intensity and female rejection; experiment 1). We fitted generalized linear models (GLMs) with temperature, mating system, and their interaction as fixed effects. Graphical inspection of LRS, actuarial aging, and reproductive behaviors (courtship intensity and male-male aggression) revealed that the normality assumption was apparently violated, as well as the independence assumption for LRS. Box–Cox transformation (**Quinn and Keough, 2002**) solved these problems and allowed us to run a GLM with a Gaussian error distribution. We compared GLMs with their corresponding null GLMs using the likelihood ratio test only to test the significance of the independent variables in the full model. We detected collinearity between the mating system and the interaction in LRS, early reproductive, reproductive aging, actuarial aging, courtship intensity, and female rejection models. In all these cases, we thus refitted the model without the main mating system effect (which was not our main interest). As a complementary analysis for LRS, we also ran a model

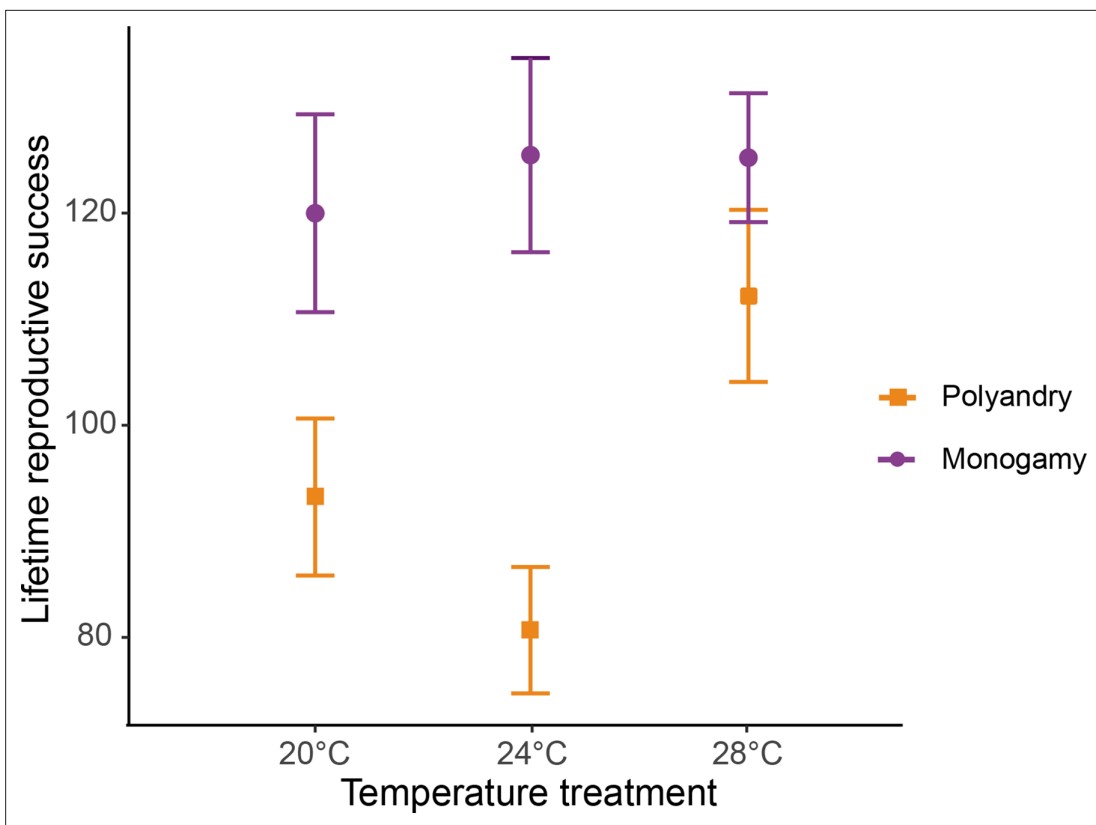

**Figure 2.** Female lifetime reproductive success (mean ± SEM) across temperature and mating system treatments. 20°C: $n_{polyandry}$ = 73 and $n_{monogamy}$ = 74. 24°C: $n_{polyandry}$ = 71 and $n_{monogamy}$ = 74. 28°C: $n_{polyandry}$ = 66 and $n_{monogamy}$ = 71.

The online version of this article includes the following figure supplement(s) for figure 2:

**Figure supplement 1.** Violin plot for female reproductive success across temperature and mating system treatments.

**Figure supplement 2.** Early reproductive rate (number of offspring produced during the first two weeks of age), late reproductive rate (number of offspring produced during the second two weeks of age), and reproductive aging (number of offspring produced over weeks 1–2 vs. 3-4) plots.

with temperature as a factor and a predetermined quadratic contrast table (given the relationship between LRS and temperature is clearly non-linear, *Figure 2*), and obtained similar results.

We also used Cox proportional hazards survival model to analyze potential differences in mortality risk across treatments, using the *survival* and *survminer* packages (*Therneau, 2015*; *Kassambara and Kosinski, 2018*), which allowed us to include the females lost during manipulations as 'right censored' individuals (i.e. individuals that are taken into account for demographic analysis until the day they disappear, *Kleinbaum and Klein, 2012*). We analyzed female rejection behaviors in two different ways. First, we examined mating system and temperature effects on overall rejection rates using a Gamma distribution, as this variable was continuous and zero-inflated. Second, we also used a binomial GLM to estimate potential differences in female rejections per courtship. Finally, we analyzed male-male aggression with the mating system as the sole main factor (as male-male aggressions where only possible in the polyandry treatment).

## Experiments 2–5

To examine the effect of temperature on post-copulatory effects, we evaluated the effect of SCR, temperature, treatment duration, and their interaction on receptivity (mating duration and remating latency -experiments 2 and 3) and fecundity -oviposition and egg viability- and female survival (experiments 4 and 5). For mating duration, remating latency and egg viability we fitted generalized linear models (GLMs) with temperature, SCR level, treatment duration, and their interaction as fixed effects. We assessed the significance of factors by dropping individual terms from the full model using the 'drop1' function, refitting models without the triple interaction where necessary. We detected a problem of collinearity between SCR and the interactions, as well as between treatment duration and the interactions in mating duration, remating latency, and egg viability models. In all these cases, we refitted the model without the main SCR and treatment duration effects (which were not our main interest). For the mating duration, we used a Gamma distribution. For remating latency and egg fertility, we used a Gaussian distribution. For oviposition, we fitted a generalized linear mixed model (GLMM) with temperature, SCR level, treatment duration and their interaction as fixed effects and day as a random effect. Initially, we run a model with a zero-inflated distribution, in which the zero values are modeled separately from the non-zero values (*Zuur et al., 2010*). However, we detected problems of collinearity, including treatment duration as an effect. We thus run two separate models for each treatment duration using Hurdle models without the main SCR effect. Finally, for survival, we used a Cox proportional hazards survival model to analyze potential differences in mortality risk across treatments, including the females lost during manipulations as 'right censored' individuals.

**Table 1.** Output from separate generalized linear models (GLMs) for each temperature level to explore significant interactions between temperature and mating system effects on female fitness components.

| | LRS | | | Reproductive aging | | | Actuarial aging | | |
|---|---|---|---|---|---|---|---|---|---|
| $T°C$ | $F_{df}$ | p-value | Estimate (95% CI) | $F_{df}$ | p-value | Estimate (95% CI) | $F_{df}$ | p-value | Estimate (95% CI) |
| 20° | $4.4_{1,145}$ | 0.039 | 1.07 (0.06–2.06) | $12.1_{1,145}$ | <0.001 | −7.99 (−12.5--3.5) | $39.6_{1,148}$ | <0.001 | 7.44 (5.1–9.8) |
| 24° | $16.6_{1,142}$ | <0.001 | 22.39 (11.6–33.1) | $35.3_{1,142}$ | <0.001 | −17.2 (−22.9-−11.5) | $32.2_{1,143}$ | <0.001 | 4.84 (3.2–6.5) |
| 28° | $2.2_{1,135}$ | 0.137 | 1.88 (−0.58–4.36) | $14.1_{1,135}$ | <0.001 | −11.87 (−18.1- -5.7) | $19.7_{1,137}$ | <0.001 | 2.97 (1.7–4.3) |

The online version of this article includes the following source data for table 1:

**Source data 1.** Summary statistics from Tukey's post hoc test to examine the meaning of significant interactions between temperature and mating system effects.

**Source data 2.** Summary statistics from Cox PH survival models as a complementary analysis to examine potential differences in mortality risk across treatments from the experiment 1.

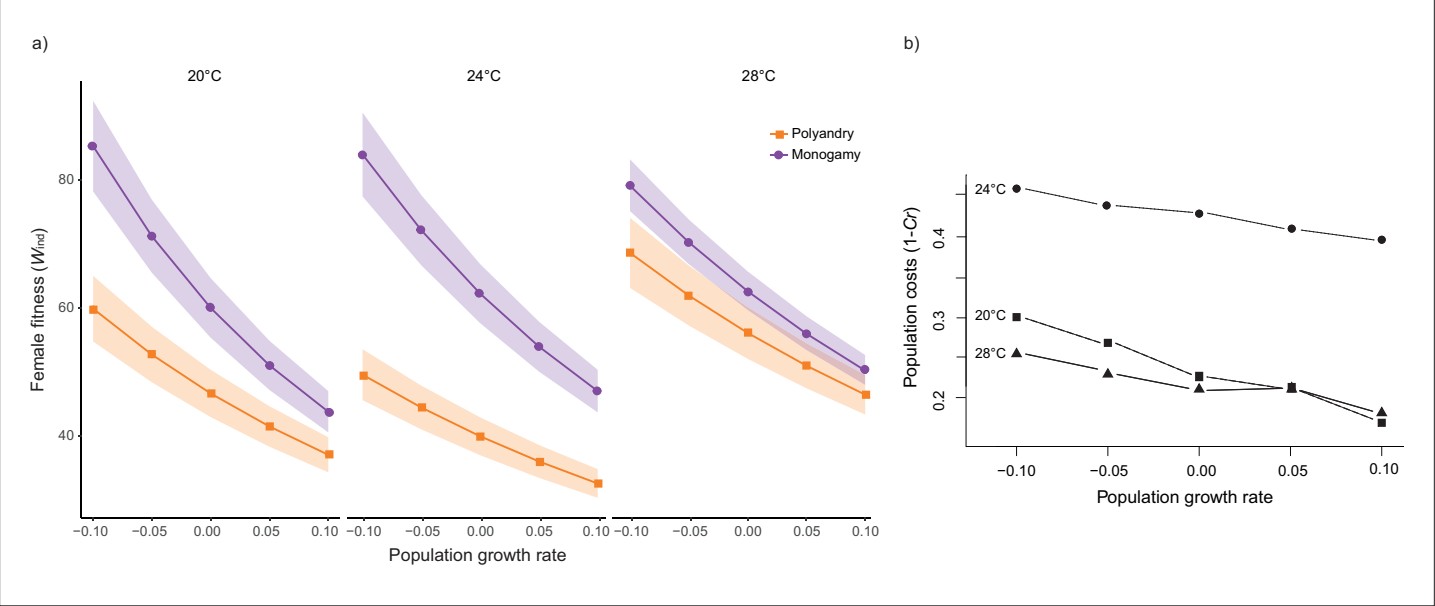

**Figure 3.** Rate-sensitive fitness estimates. (**a**) Average rate-sensitive index fitness estimate of individual females (Mean $\omega_{ind}$) for different population growth rates across temperature and mating system treatments (shaded areas denote SEM). 20°C: $n_{polyandry}$ = 73 and $n_{monogamy}$ = 74. 24°C: $n_{polyandry}$ = 71 and $n_{monogamy}$ = 74. 28°C: $n_{polyandry}$ = 66 and $n_{monogamy}$ = 71. (**b**) Relative cost ($C_r$) of polyandry (vs. monogamy) for each temperature treatment for different population growth rates. $Cr$ was calculated based on rate-sensitive index fitness estimates for populations ($\omega_{pop}$), whereby population costs are shown as 1 – $Cr$, thus reflecting the relative decrease in population growth rate.

## Results

### Net impact of male harm on female fitness and underlying behavioral mechanisms (experiment 1)

#### LRS

We detected a significant temperature by mating system interaction for female lifetime reproductive success ($F_{1,425}$ = 16.931, p<0.001, *Figure 2*, *Figure 2—figure supplement 1*), and a marginally non-significant temperature effect ($F_{1,425}$ = 3.712; p=0.054). Separate models for each temperature level show a larger effect of the mating system on lifetime reproductive success at 24°C than at 20°C, and larger at 20°C than at 28°C, despite 95 % CI of the estimates overlaps (*Table 1*, *Table 1—source data 1a*). The decrease in LRS in polyandry vs. monogamy peaked at 24°C (H=0.36) and was 1.6 (H=0.22) and 3.4 times (H=0.10) smaller at 20°C and 28°C, respectively. Rate-sensitive fitness estimates show that estimated population costs are dependent on background growth rates (*Figure 3*), and in general particularly accused in decreasing populations.

#### Early reproductive rate

We did not detect a significant temperature by mating system interaction ($F_{1,425}$ = 2.94; p=0.09). We did detect a significant main temperature effect ($F_{1,425}$ = 64.63; p<0.001; *Figure 2—figure supplement 2*), such that early reproduction increased at 28°C.

#### Reproductive aging

We detected a significant temperature by mating system interaction for reproductive aging ($F_{1,425}$ = 55.24; p<0.001; *Figure 2—figure supplement 2*), and a clear main effect for temperature ($F_{1,425}$ = 44.25; p<0.001). Running models separately for each temperature level showed that mating system affected reproductive aging at all temperatures, but particularly so at 24°C and 28°C (*Table 1*, *Table 1—source data 1a*, *Figure 2—figure supplement 2*).

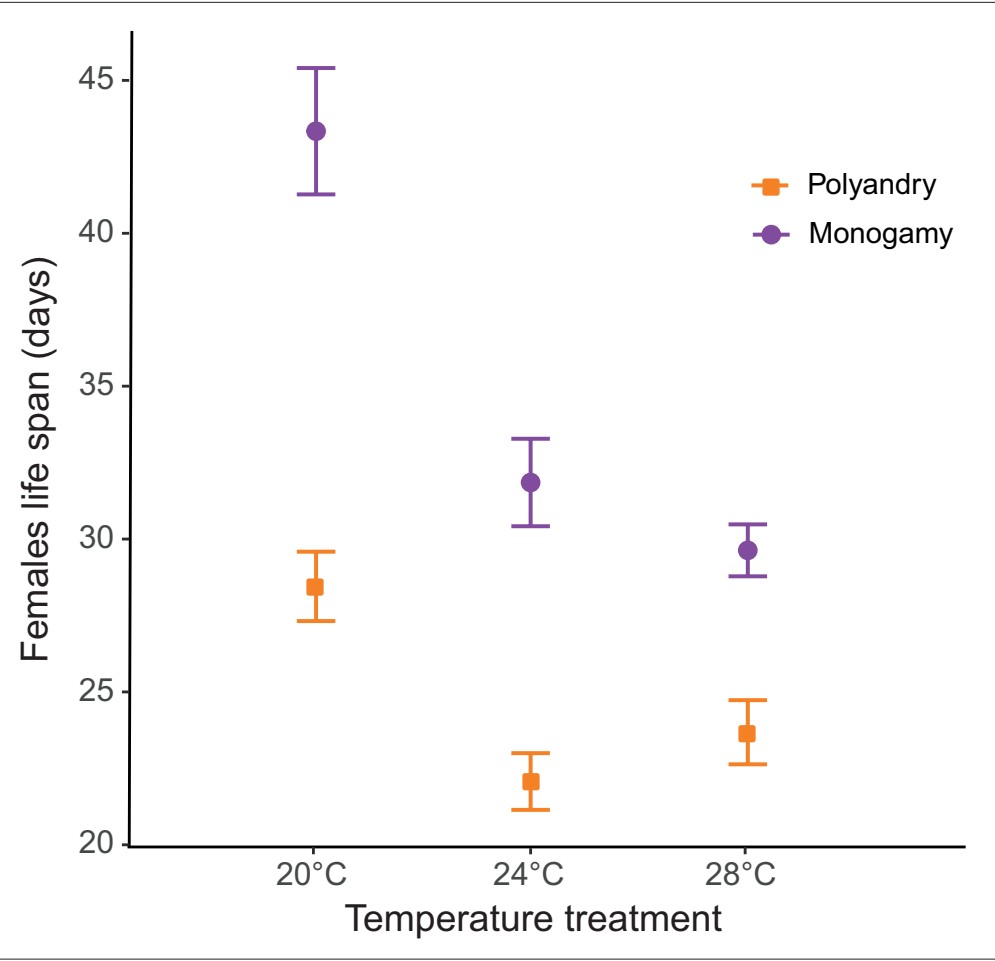

**Figure 4.** Male harm effect on female lifespan (mean ± SEM) across temperature and mating system treatments. 20°C: $n_{polyandry}$ = 73 and $n_{monogamy}$ = 74. 24°C: $n_{polyandry}$ = 71 and $n_{monogamy}$ = 73. 28°C: $n_{polyandry}$ = 66 and $n_{monogamy}$ = 73. The online version of this article includes the following figure supplement(s) for figure 4:

**Figure supplement 1.** Male harm effect on female lifespan across temperature and mating system treatments.

### Actuarial aging

We detected a significant temperature by mating system effect for lifespan ($F_{1,428}$ = 73.81; p<0.001; *Figure 4*, *Figure 4—figure supplement 1a*), and a significant main effect for temperature ($F_{1,428}$ = 36.98; p<0.001). Mating system affected actuarial aging at all temperatures, but particularly so at 20°C (*Table 1*, *Table 1—source data 1*). The survival analysis yielded qualitatively identical results (*Table 1—source data 2*, survival plot *Figure 4—figure supplement 1b*).

### Reproductive behaviour

The interaction between temperature and mating system was significant for courtship rate ($F_{1,441}$ = 45.62; p<0.001), and we also detected a main temperature effect ($F_{1,441}$ = 16.69; p<0.001 *Figure 5a*, *Figure 5—figure supplement 1a*). Running models separately for each temperature level, mating system affected courtship rate at 24°C and 28°C but not at 20°C (*Table 2*, *Table 1—source data 2b*). Likewise, we detected a significant temperature by mating system effect for rejection rate ($F_{1,441}$ = 24.48; p<0.001 *Figure 5b*, *Figure 5—figure supplement 1b*), and a main effect for temperature ($F_{1,441}$ = 5.61; p=0.020). Models for each temperature level show a mating system effect at 24°C and 28°C but not at 20°C (*Table 2*, *Table 1—source data 2b*). We did not detect a significant interaction between temperature and mating system ($F_{1,294}$ = 0.30; p=0.582), nor a temperature effect ($F_{1,294}$=0.08; p=0.773), for rejection rates per courtship. For male-male aggression rate, we detected a significant temperature effect ($F_{1,214}$ = 14.45; p<0.001 *Figure 5c*, *Figure 5—figure supplement 1c*).

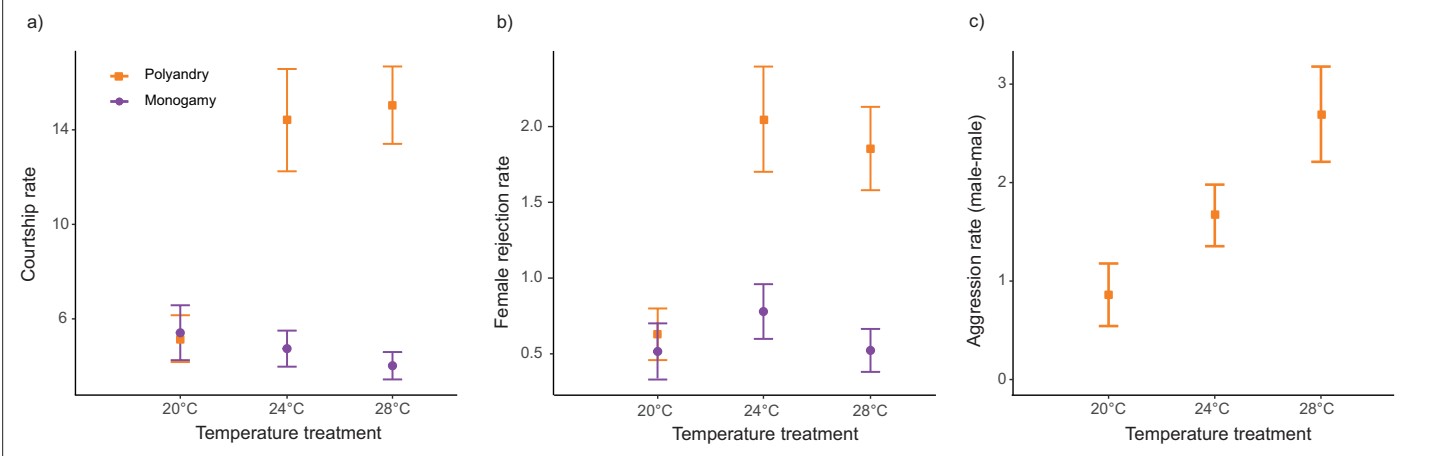

**Figure 5.** Reproductive behaviors (mean ± SEM) across temperature and mating system treatments. (**a**) Courtships per female per hour, (**b**) Female rejections per hour, and (**c**) Aggressions male-male per hour. 20°C: $n_{polyandry}$ = 74 and $n_{monogamy}$ = 76. 24°C: $n_{polyandry}$ = 72 and $n_{monogamy}$ = 77. 28°C: $n_{polyandry}$ = 70 and $n_{monogamy}$ = 75.

The online version of this article includes the following figure supplement(s) for figure 5:

**Figure supplement 1.** Violin plot for male harm effect on: (**a**) Courtship rate and (**b**) Rejection rate across temperature and mating system treatments; (**c**) Violin plot for polyandry mating system effect on aggression rate.

**Figure supplement 2.** Total number of matings across the 8 hr of observations.

## Mating effects on female reproduction and survival (experiments 2 to 5)

### Female receptivity (experiments 2 and 3)

For the duration of the first mating in our female receptivity assays, we detected significant SCR by temperature ($F_{1,1239}$ = 40.42; p<0.001), treatment duration by temperature ($F_{1,1239}$ = 5.97; p=0.024), and SCR by treatment duration ($F_{1,1239}$ = 5.48; p=0.024) interactions (*Figure 6a*, *Figure 6—figure supplement 1a*). We found no significant main effect for temperature ($F_{1,1239}$ = 3.36; p=0.07). Running models separately for each temperature, SCR affected the duration of the first mating at 20°C, 24°C, and 28°C, while treatment duration only affected the duration of the first mating at 24°C (*Table 3a*, *Table 3—source data 1a-b*). Additionally, running models for each treatment duration, SCR affected the duration of the first mating at short and long treatment durations (*Table 3b*, *Table 3—source data 1c*). Similarly, for female remating latency, we detected significant interactions for SCR by temperature ($F_{1,1094}$ = 6.15; p=0.022), and treatment duration by temperature ($F_{1,1094}$ = 5.17; p=0.028), whereas the interaction between SCR by treatment duration was not significant ($F_{1,1094}$ = 1.00; p=0.316) (*Figure 6b*, *Figure 6—figure supplement 1b*). We also detected a significant main temperature effect ($F_{1,1094}$ = 8.21; p=0.01). Running models separately for each temperature level, SCR level, and treatment duration only affected remating latency at 28°C (*Table 3a*, *Table 3—source data 1a-b*).

**Table 2.** Output from separate generalized linear models (GLMs) for each temperature level to explore significant interactions between temperature and mating system effects on underlying behavioural mechanisms.

p-values were corrected for multiple testing using Benjamini-Hochberg correction.

| $T°C$ | Courtship rate | | | Rejection rate | | |
|---|---|---|---|---|---|---|
| | $F_{df}$ | *p-value* | *Estimate (95% CI)* | $F_{df}$ | *p-value* | *Estimate (95% CI)* |
| 20° | $0.4_{1,148}$ | 0.546 | −0.04 (−0.16–0.08) | $0.20_{1,148}$ | 0.654 | −0.05 (−0.30–0.19) |
| 24° | $21.8_{1,147}$ | **<0.001** | −0.40 (−0.57- -0.23) | $10.9_{1,147}$ | **0.001** | −17.2 (−1.01- -0.25) |
| 28° | $40.2_{1,143}$ | **<0.001** | −0.63 (−0.83- -0.43) | $19.3_{1,143}$ | **<0.001** | −11.87 (−0.96- -0.36) |

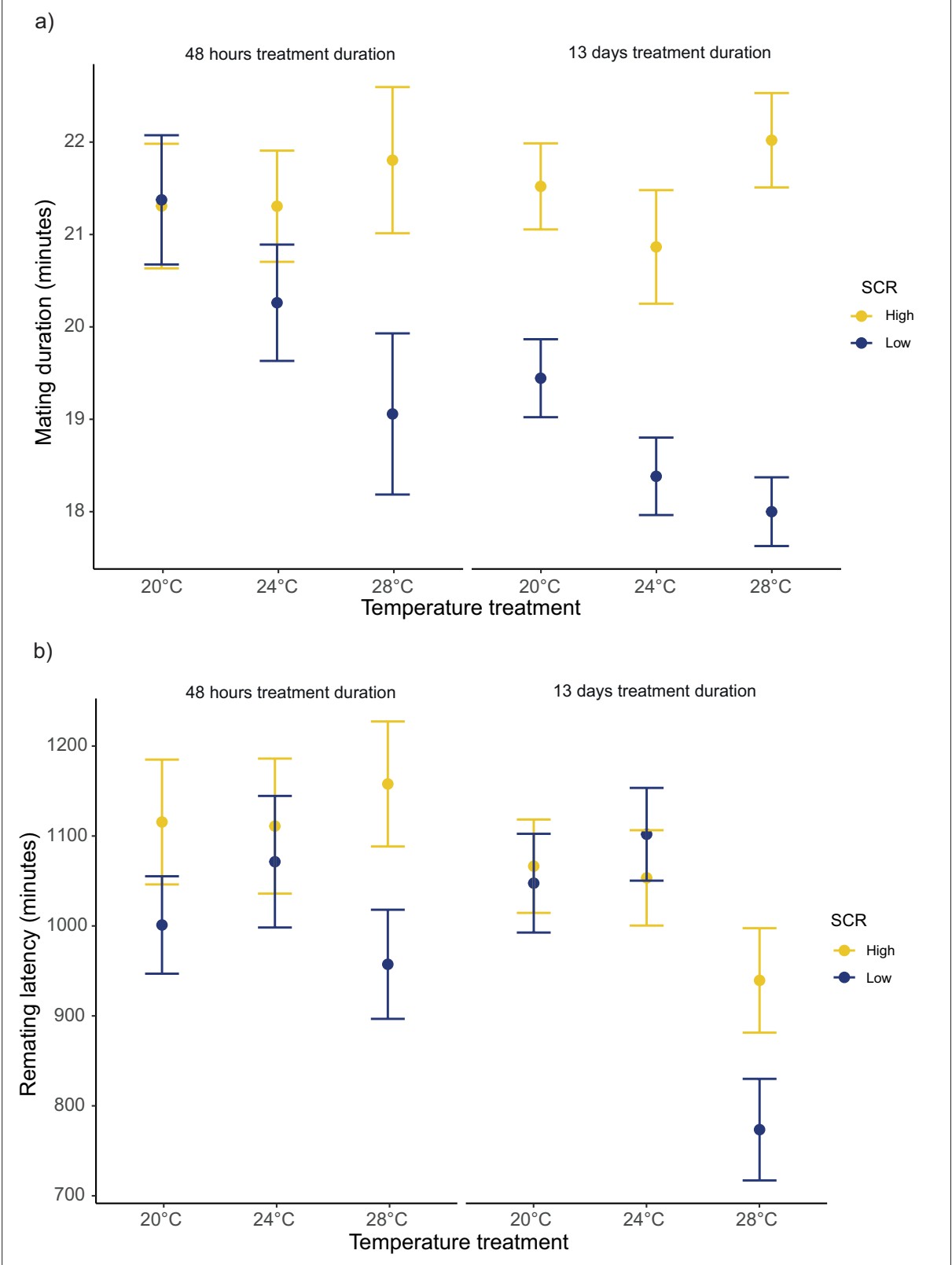

**Figure 6.** Mean ± SEM for mating duration and remating latency. (**a**) Mating duration of males exposed to high (8 males per vial) or low sperm competition risk (1 male per vial) for 48 hr or 13 days prior to mating at different temperatures. 20°C: $n_{high/48hr}$ = 91, $n_{low/48hr}$ = 96, $n_{high/13days}$ = 121 and $n_{low/13days}$ = 117. 24°C: $n_{high/48hr}$ = 85, $n_{low/48hr}$ = 88, $n_{high/13days}$ = 119 and $n_{low/13days}$ = 115. 28°C: $n_{high/48hr}$ = 92, $n_{low/48hr}$ = 104, $n_{high/13days}$ = 99 and $n_{low/13days}$ = 117. (**b**) Female remating latency following a single mating with either a male from a high or low sperm competition risk level for 48 hr or 13 days before

*Figure 6 continued on next page*

*Figure 6 continued*

mating across temperature treatments. 20°C: $n_{high/48hr}$ = 75, $n_{low/48hr}$ = 73, $n_{high/13days}$ = 119 and $n_{low/13days}$ = 113. 24°C: $n_{high/48hr}$ = 61, $n_{low/48hr}$ = 70, $n_{high/13days}$ = 116 and $n_{low/13days}$ = 113. 28°C: $n_{high/48hr}$ = 63, $n_{low/48hr}$ = 82, $n_{high/13days}$ = 98 and $n_{low/13days}$ = 117.

The online version of this article includes the following figure supplement(s) for figure 6:

**Figure supplement 1.** Violin plots for (**a**) Mating duration of males exposed to a high (8 males per vial) or low sperm competition risk (1 male per vial) level 48 hr (experiment 2) and 13 days (experiment 3) before mating across temperature treatments, and (**b**) Female remating latency following a single mating with either a male from a high or low sperm competition risk level, for both 48 hr and 13 days of temperature treatment duration before mating in a common garden.

**Figure supplement 2.** Eggs produced by females during the first three days following a single mating with treated males.

**Figure supplement 3.** Total of offspring produced by females during the days 1, 2, 3, 4, 5, and 8 after mating following a single mating with treated males.

**Figure supplement 4.** Female lifespan after mating following a single mating with treated males.

## Female fecundity and survival (experiments 4 and 5)

For the number of eggs produced by females during the three first days, we did not detect significant interactions between temperature and SCR for either short or long treatment durations, nor a main significant effect for temperature (*Figure 6—figure supplement 2* & *Table 4—source data 1*): (i) Short treatment duration, SCR by temperature interaction ($\chi^2_1$=0.05; p=0.821), temperature effect ($\chi^2_1$=2.82; p=0.092), (ii) Long treatment duration, SCR by temperature interaction ($\chi^2_1$=0.03; p=0.840), temperature effect ($\chi^2_1$=0.37; p=0.541). Likewise, for the total number of offspring produced by females during days 1, 2, 3, 4, 5, and 8 after mating, we did not find significant interactions between SCR and treatment duration ($F_{1,952}$ = 0.022; p=0.881), or between SCR and temperature ($F_{1,952}$ = 0.418; p=0.674), but we did between temperature and treatment duration ($F_{1,952}$ = 9.599; p=0.005)

**Table 3.** Model outputs from separate generalized linear models (GLMs) for each (a) temperature level and (b) treatment duration to explore significant interactions.

**a)**

| T°C | Effect | Mating duration | | | Remating latency | | |
|---|---|---|---|---|---|---|---|
| | | $F_{df}$ | p-value | Estimate (95% CI) | $F_{df}$ | p-value | Estimate (95% CI) |
| | Sperm competition risk | $3.9_{1,423}$ | **0.046** | 0.03 (0.0005– 0.05) | $0.95_{1,377}$ | 0.330 | 27.9 (−28.2– 84.2) |
| 20° | Treatment duration | $2.3_{1,423}$ | 0.133 | −0.02 (−0.04– 0.006) | $0.0006_{1,377}$ | 0.980 | −0.73 (−58.5– 57.0) |
| | Sperm competition risk | $10.6_{1,405}$ | **0.001** | 0.05 (0.02– 0.07) | $0.07_{1,358}$ | 0.779 | −8.47 (−67.7– 50.7) |
| 24° | Treatment duration | $3.7_{1,405}$ | **0.054** | −0.02 (−0.05– 0.0003) | $0.04_{1,358}$ | 0.842 | −6.24 (−67.8– 55.3) |
| | Sperm competition risk | $26.5_{1,410}$ | **<0.001** | 0.084 (0.052– 0.117) | $8.05_{1,358}$ | **0.005** | 87.81 (27.1– 148.4) |
| 28° | Treatment duration | $0.6_{1,410}$ | 0.451 | −0.12 (−0.04- -0.2) | $9.73_{1,358}$ | **0.002** | −97.65 (−158.9- -36.3) |

**b)**

| Treatment duration | Mating duration | | |
|---|---|---|---|
| | $F_{df}$ | p-value | Estimate (95% CI) |
| Short (48 hr) | $4.5_{1,554}$ | **0.033** | 0.03 (0.002– 0.06) |
| Long (13 days) | $54.2_{1,686}$ | **<0.001** | 0.07 (0.051– 0.089) |

The online version of this article includes the following source data for table 3:

**Source data 1.** Summary statistics from Tukey's post hoc test as a complementary analysis to examine the meaning of significant interactions found for mating duration and remating latency.

**Table 4.** Summary statistics from fitting generalized linear models (GLMs) separately for each temperature level to explore the significant interaction between temperature and treatment duration effects for total offspring produced by females during days 1, 2, 3, 4, 5, and 8 after mating.

| | Total of offspring | | |
|---|---|---|---|
| $T°C$ | $F_{df}$ | p-value | Estimate (95% CI) |
| 20° | $0.6_{1,322}$ | 0.454 | 1.42 (−2.30–5.15) |
| 24° | $4.6_{1,321}$ | 0.032 | 4.11 (0.35–7.86) |
| 28° | $5.2_{1,308}$ | 0.022 | 4.26 (0.62–7.89) |

The online version of this article includes the following source data for table 4:

**Source data 1.** Summary statistics from the Hurdle model to analyze potential differences in egg production across treatments with temperature as a factor.

**Source data 2.** Summary statistics from Tukey's post hoc test as a complementary analysis to examine the meaning of significant interaction between temperature and treatment duration for total of offspring produced by females during the days 1, 2, 3, 4, 5, and 8 after mating.

(*Figure 6—figure supplement 3*). We did not detect a main temperature effect ($F_{1,952}$ = 2.797; p=0.157). Running models for each temperature level, we found that treatment duration affected the total number of offspring produced by females at 24°C and 28°C, but not at 20°C (*Table 4*, *Table 4—source data 2*).

Finally, we did not detect significant interactions in survival (SCR by treatment duration, $\chi^2_1$=0.276, p=0.694; SCR by temperature, $\chi^2_1$=0.311, p=0.694; temperature by treatment duration, $\chi^2_1$=4.09, p=0.128), nor significant main temperature ($\chi^2_1$=1.69; p=0.386) or SCR effects ($\chi^2_1$=0.154; p=0.694). We did detect a significant main treatment duration effect ($\chi^2_1$=13.03; p=0.001; *Figure 6—figure supplement 4*).

## Discussion

We show that male harm exhibits complex plasticity in response to temperature changes within an optimal reproductive range (20–28°C) for a wild *D. melanogaster* population, with several implications for our understanding of how sexual conflict unfolds in nature. First, we show that net harm to females varies markedly within this range, such that relative harm (i.e. the proportional reduction in average female LRS in polygamy vs. monogamy) was maximal at 24°C, decreased at 20°C and was minimum at 28°C. Second, rate-sensitive fitness estimates indicate that arising population costs are dependent on the interaction between temperature and population demography, such that demography modulates the impact of male harm on population viability less at warmer temperatures. Third, our results strongly suggest that male harm effects on population growth have to do with the fact that different mechanisms exhibit qualitatively different reaction norms in response to temperature, with distinct effects on different female fitness components. More specifically, at cold temperatures courtship intensity (i.e. male harassment; pre-copulatory harm mechanism) decreased, and female fitness was impacted more via accelerated actuarial aging than at warm temperatures. In contrast, warm temperatures impacted mating costs and effects on female receptivity (i.e. post-copulatory mechanism), and female fitness decreased more via accelerated reproductive aging than at colder temperatures. We discuss how such plasticity may affect how male harm impacts populations, as well as selection, adaptation and, ultimately, evolutionary rescue under a warming climate.

### Temperature effects on male harm and its consequences for populations

We found that temperature variation within the optimal reproductive range for our study population in the wild had a strong effect on net male harm levels. To gauge male harm, we used the standard procedure of comparing female LRS in monogamy, which imposes low male-male competition and thus low sexual conflict, vs. polyandry (i.e. a female with three males), which imposes high male-male competition and intensifies sexual conflict between the sexes (*Yun et al., 2021*). These sex ratios are common in mating patches in the wild, and are actually representative of the extremes in natural levels of male-male competition (*Dukas, 2020*). In monogamy, temperature did not affect female fitness (*Figure 2*), showing that female reproduction is indeed optimal within this range. In contrast, the net decrease in female LRS in polyandry vs. monogamy was highly dependent on the thermal environment, with an average decrease of H=0.36 at 24 °C, H=0.22 at 20 °C. and H=0.10 at 28 °C, at which temperature we did not find a statistically significant effect of mating system on female LRS (*Figure 2*).

Male harm effects are expected to be cumulative over time, so that their impact on female survival and reproductive output is unlikely to be constant across a female's lifespan (*Bondurianky et al., 2008*; *Filice et al., 2020*). At the same time, early vs. late-life reproduction weigh differently on both individual fitness and how this impacts background population growth, depending on whether such population is decreasing, stable, or growing (*Edward et al., 2011*; *Priest et al., 2008*). Thus, in order to evaluate how male harm is likely to impact populations across different temperatures, we calculated rate-sensitive fitness estimates for individual (*Figure 3a*) and population (*Figure 3b*) fitness across a range of demographic scenarios (i.e. decreasing, stable, and growing populations; *Edward et al., 2011*). Overall, the impact of male harm was higher in decreasing populations, where late-life reproduction gains importance, which is consistent with the idea that male harm effects are cumulative over the lifespan of females. Above and beyond this general effect, the observed interaction between temperature and female fitness is maintained irrespective of population demography. That is, male harm decreases female individual fitness more at 24°C than at 20°C and 28°C. Interestingly, though, temperature also has a clear effect on how demography affects population-level costs. At hotter temperatures the relative population costs of male harm vary considerably less with demography (i.e. background population growth) than at colder temperatures (*Figure 3b*). For a decreasing population, the relative population costs of male harm are significantly higher at 20°C than at 28°C, but this difference wanes as population growth rate (r) increases, to the point of reverting in a rapidly growing population (*Figure 3b*). This suggests that male harm has more impact on late-life female fitness at cold temperatures, and hints at the possibility that cold vs. hot temperature affect qualitatively distinct parameters of female fitness, and thus underlying mechanisms of male harm.

Looking at the effects on separate female fitness components yields results largely in agreement with the above ideas. Again, consistent with the fact that harm needs to accumulate in time to impact female fitness, temperature had no effect on how or whether male competition impacted early reproductive rate (*Figure 2—figure supplement 2*). Temperature did, however, modulate how male harm impacted on female actuarial vs. reproductive aging. We found clear differences in female lifespan across temperature and mating system treatments. Male harm effects on actuarial aging (an increase in mortality rate with age) were more severe at 20°C (35% decrease in female lifespan) than at 24°C (31% decrease) and at 28°C (22% decrease; *Figure 3*). In contrast, while male harm accelerated reproductive aging at all temperatures, this decrease was more marked at 24°C and at 28°C than at 20°C (*Table 1*; *Figure 2—figure supplement 2*). In accordance with available evidence in lab-adapted flies (*García-Roa et al., 2019*), these results show that temperature does not seem to have a linear effect on aging processes.

To sum up, we offer strong evidence that different male harm mechanisms are sensitive to temperature in different ways, with ensuing modulation of its effects on different female fitness components. We underscore two potential consequences arising from these findings. The first is that the net fitness effects of male harm might be lower than expected when considered in its natural thermal setting because previous research has focused on studying male harm at average temperatures, precisely where we found it to be maximal. It follows that integrating harm across the natural temperature range will result in a lower net decrease in female fitness in the wild. The second is that environmental variability may foster the maintenance of genetic variation underlying different mechanisms of male harm, and even potentially divergent male-male competition strategies. Our work joins an increasing number of recent studies in highlighting the importance of evaluating more ecologically realistic scenarios in sexual conflict research, particularly how natural fluctuations in the ecology of the socio-sexual context may affect sexual conflict processes (*García-Roa et al., 2019*, *García-Roa et al., 2020*; *Gomez-Llano et al., 2018*; *Yun et al., 2017*).

In addition, the above results open up the possibility that warm climates may lessen the impact of sexual conflict on population viability, perhaps facilitating evolutionary rescue. Male harm effects were found to be relatively lower in warmer temperatures and in decreasing populations, precisely the type of context that would be typical of a climate-change scenario. The effects found in this study were within the optimum reproductive range for this population, but similar results have been reported in response to stressful temperatures. For instance, temperature has been used to induce environmental stress in natural populations of seed beetles (*Callosobruchus maculatus*), showing that a stressful thermal regime reduces intra-locus sexual conflict by aligning selection in males and females (*Berger et al., 2014*, but see *Martinossi-Allibert et al., 2019*). A previous study in a lab-adapted population

of *D. melanogaster* also shows that male harm levels (i.e. inter-sexual conflict) decrease when subject to maladaptive warm temperatures (*García-Roa et al., 2019*). However, there are two reasons why it is important to caution against direct extrapolation of our results to wild populations at this stage, in particular in relation to their relevance for populations facing the current climate crisis. First, our monogamy vs. polyandry treatments reflect the low vs. high-end of the spectrum of operational sex ratios that are typical of *D. melanogaster* at mating patches in the wild (*Dukas, 2020*), and thus our measures of male harm effects are likely to be generally higher than expected in nature. While this does not change the main conclusions regarding temperature effects, it is important to note when considering the degree to which these effects may be relevant in the wild. Second, our treatment temperatures were stable, designed to study how coarse-grain changes in temperature across the adult lifespan of flies may influence how sexual conflict unfolds in nature. Thus, future studies will need to encompass how fine-grained fluctuation (i.e. repeated variation of temperature across an adult's lifespan) may affect male harm for a more comprehensive picture of temperature effects on sexual conflict in the wild.

## Temperature effects on sexual conflict mechanisms in *Drosophila melanogaster*

Prior to mating, *D. melanogaster* males harm females via sexual harassment, due to protracted court-ship of one or several males that results in physical damage, interference with other behaviors (e.g. foraging or egg-laying), and costly energetic investment into male avoidance (e.g. female rejection) (*Bretman et al., 2019*; *Partridge and Fowler, 1990*; *Teseo et al., 2016*). Importantly, previous studies in this species have shown that male harm is directly related to courtship intensity and female rejection, and indirectly to male-male aggression as a direct measure of male intrasexual competition (e.g. *Bretman et al., 2019*; *Carazo et al., 2014*; *Partridge and Fowler, 1990*). In our study, we found a clear increase in both courtship intensity and female rejection in polyandry vs. monogamy, but this was largely dependent on the thermal environment (*Figure 5a–b*). While we found clear evidence that harassment increases in polygamy at 28°C and 24°C, we did not find a similar effect at 20°C (*Figure 5a*). Female rejection behaviors exhibited the same trend (*Figure 5b*), and this was due to increased male courtship attempts (not to an increase in female likelihood to reject male courtships). Thus, our results suggest that male harassment decreases drastically at cold temperatures and is perhaps maximal at warm temperatures, at which temperatures we also detected the highest level of male-male aggression (*Figure 5c*). The above results seem to suggest that the decrease in male harm to females that we detected at 20°C vs. 24°C (*Figure 2*) is likely explained by the substantial drop in male harassment at this temperature. However, the same logic cannot apply to the decrease in male harm to females that we detected at 28°C vs. 24°C (see below).

During mating, *D. melanogaster* males transfer seminal fluid proteins (SFPs) that manipulate female re-mating and egg-laying rates to their advantage, but this normally comes at a cost to females in terms of lifespan and lifetime reproductive success (*Chapman et al., 2003b*; *Chapman et al., 1995*; *Sirot et al., 2009*; *Wigby and Chapman, 2005*). Furthermore, these effects are known to be modulated by the socio-sexual context, such that males strategically adjust their investment in SFPs depending on expected SCR levels (*Bretman et al., 2009*; *Hopkins et al., 2019*; *Wigby et al., 2009*). We run a series of standard assays in *Drosophila* (*Bretman et al., 2009*; *Hopkins et al., 2019*; *Wigby et al., 2009*) to investigate whether temperature modulates any of the known phenotypic effects that SFPs have on females following a single mating with males that were exposed to low (i.e. kept alone in their vial) or high SCR (i.e. kept with seven other males in a vial). The temperature males were kept at prior to mating did not modulate how mating affected short-term female fecundity or lifespan, but we did detect a clear effect of temperature with respect to female receptivity. Generally, reception of a male ejaculate resulted in a sharper decrease in female receptivity (i.e. longer remating latency) when males perceived a high SCR, which is to be expected and is in accordance with previous studies (*Bretman et al., 2009*; *Bretman et al., 2010*; *Denis et al., 2017*). However, this effect was largely modulated by temperature, such that it was more clearly detected at 28°C (*Figure 6b*). Interestingly, this effect was consistent for males treated for 48 hr and 13 days (albeit more marked in the latter case), and these results were paralleled by temperature effects on mating duration (*Figure 6a*). Namely, males exposed to high SCR generally mated for longer than males exposed to low SCR, again in line with previous findings (*Bretman et al., 2010*; *Bretman et al., 2013*), but this difference was

clearly larger at 28°C (*Figure 6a*). It is worth restating that, under monogamous conditions, we did not detect a decrease in female productivity at 28°C vs. 24°C and 20°C, so that general differences in sperm viability are unlikely to account for the aforementioned differences. A potential explanation for these results is that males may perceive a higher SCR when kept in groups at warmer temperatures (e.g. due to increased activity). This, however, would predict the same differences between 24°C and 20°C, which was not the case.

All in all, our results suggest that at least some post-copulatory harm mechanisms are sensitive to temperature, because receipt of a male ejaculate resulted in a sharper decrease of female receptivity in high vs. low SCR at warm temperatures (particularly after 13 days of exposure). We speculate that this may contribute to explain why male harm drops so sharply at 28°C despite the fact that male harassment and male-male competition seem to be maximal at this temperature. In *D. melanogaster*, as in many other species, repeatedly mating is costly for females in terms of lifetime reproductive success (*Arnqvist and Nilsson, 2000*; *Chapman et al., 1995*; *Fowler and Partridge, 1989*). An intriguing possibility is thus that either SFPs are more effective at lowering female re-mating rates or males are investing relatively more in SFPs with increasing SCR at warm temperatures, thereby buffering these costs. An alternative (but complementary) possibility is that temperature may affect female behaviour or physiology in a way that makes them more resistant to harm.

We suggest future studies should explore these ideas by examining in detail how temperature affects the composition and transfer of SFPs to females, and how females respond to the transfer of these proteins and to male harm in general (i.e. effects on female resistance). In combination with experimental evolution at different temperatures, such an approach would allow us to disentangle between two causal hypotheses for the observed results. First, that warm temperatures may buffer sexual conflict in itself by aligning male and female reproductive interests. For example, if live-fast-die-young strategies fare relatively better for females at warm than cold temperatures, male and female optimal reproductive strategies may overlap more due to the fact that cumulative late-life effects of male harm might be diluted by the inherently high female intrinsic mortality at warm temperatures. Second, whether modulation of male harm at cold and (particularly) warm temperatures has to do with the fact that different male harm mechanisms are adapted to operate better at certain temperatures. For example, due to environmental effects on male activity or protein folding. In the latter case, male harm would be expected to increase as males adapt to higher or lower average temperatures, but sexual conflict per se (i.e. the degree to which male and female evolutionary interests overlap) would be expected to remain constant. Both of the above hypotheses could have broad consequences for our understanding of the evolution of sexual conflict across the tree of life.

## Conclusions

Our findings may have implications for our understanding of how sexual conflict unfolds in nature, and its consequences for populations. First, they add to growing evidence (*Gomez-Llano et al., 2018*; *MacPherson et al., 2018*; *Malek and Long, 2019*; *Perry and Rowe, 2018*; *Yun et al., 2017*) indicating that ecological context is key in shaping sexually antagonistic coevolution and, in particular, suggest that temperature may be a particularly salient ecological factor to understand how sexual conflict evolves and operates in nature (*García-Roa et al., 2020*). Second, they highlight that male harm mechanisms can be highly plastic even in response to relatively minor fluctuations in temperature well within the optimal reproductive range, and suggest that different harm mechanisms are differently affected by temperature. Third, they suggest that male harm effects on female life-history and fitness components are asymmetrically modulated by temperature; male harm particularly decreased survival at cold and moderate temperatures, and reproductive aging at moderate and hot temperatures. In conjunction, these phenomena may have a bearing on evolutionary rescue and local adaptation processes. For example, in maintaining genetic variation in sexually selected traits in males, and/or in ameliorating the demographic impact of sexual conflict in populations facing environmental change.

## Acknowledgements

We thank Alejandro Hita for assistance with the experimental procedure. PC was supported by a research grant (PID2020-118027GB-I00) funded by MCIN/AEI/ 10.13039/501100011033 and a research grant AICO/2021/113 from Generalitat Valenciana. RGR was supported by a postdoctoral grant (FJC2018-037058-I) funded by MCIN/AEI/ 10.13039/501100011033 and by a Marie Sklodowska

Curie Fellowship (HORIZON-MSCA-2021-PF-01 101061275). CLN was supported by a predoctoral grant (PRE2018-084009) by MCIN/AEI/ 10.13039/501100011033 and by "ESF Investing in your future".

## Additional information

### Funding

| Funder | Grant reference number | Author |
| --- | --- | --- |
| Ministerio de Ciencia e Innovación | PID2020-118027GB-I00 | Pau Carazo |
| Generalitat Valenciana | AICO/2021/113 | Pau Carazo |
| Ministerio de Educación y Formación Profesional | FJC2018-037058-I | Roberto García-Roa |
| Marie Sklodowska-Curie Actions | HORIZON-MSCA-2021-PF-01 101061275 | Roberto García-Roa |
| Ministerio de Asuntos Económicos y Transformación Digital, Gobierno de España | PRE2018-084009 | Claudia Londoño-Nieto |
| European Social Fund | ESF Investing in your future | Claudia Londoño-Nieto |

The funders had no role in study design, data collection and interpretation, or the decision to submit the work for publication.

### Author contributions

Claudia Londoño-Nieto, Data curation, Formal analysis, Validation, Investigation, Visualization, Methodology, Writing – original draft; Roberto García-Roa, Supervision, Investigation, Methodology, Writing - review and editing; Clara Garcia-Co, Paula González, Methodology; Pau Carazo, Conceptualization, Resources, Supervision, Funding acquisition, Validation, Investigation, Visualization, Methodology, Writing - review and editing

### Author ORCIDs

Claudia Londoño-Nieto ⓘ http://orcid.org/0000-0002-7408-7327
Roberto García-Roa ⓘ http://orcid.org/0000-0002-9568-9191
Clara Garcia-Co ⓘ http://orcid.org/0000-0002-4126-5940
Pau Carazo ⓘ http://orcid.org/0000-0002-1525-6522

### Decision letter and Author response

Decision letter https://doi.org/10.7554/eLife.84759.sa1
Author response https://doi.org/10.7554/eLife.84759.sa2

## Additional files

### Supplementary files

• MDAR checklist

### Data availability

All data generated or analysed during this study are included in the manuscript. Source data files are uploaded to Dryad repository (https://doi.org/10.5061/dryad.pzgmsbcqz), along with R script https://doi.org/10.5281/zenodo.7350587.

The following dataset was generated:

| Author(s) | Year | Dataset title | Dataset URL | Database and Identifier |
|---|---|---|---|---|
| Londoñ-Nieto C, García-Roa R, Garcia-Co C, González P, Carazo P | 2023 | Thermal phenotypic plasticity of pre- and post-copulatory male harm buffers sexual conflict in wild *Drosophila melanogaster* | https://doi.org/10.5061/dryad.pzgmsbcqz | Dryad Digital Repository, 10.5061/dryad.pzgmsbcqz |

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
