## [Editor Report]

This study has important implications for the impact of sexual conflict on population viability under different temperatures. The authors provide compelling evidence that male harm to females in sexual conflict can be reduced as a function of temperature within the optimal reproductive range of a species. The results have implications for the likelihood of the evolutionary rescue of species facing the climate crisis.

---

## [Decision Letter]

**Decision letter after peer review:**

Thank you for submitting your article "Thermal phenotypic plasticity of pre-and post-copulatory male harm buffers sexual conflict in wild *Drosophila melanogaster*" for consideration by *eLife*. Your article has been reviewed by 3 peer reviewers, and the evaluation has been overseen by a Reviewing Editor and George Perry as the Senior Editor. The following individuals involved in the review of your submission have agreed to reveal their identity: Ivain Martinossi (Reviewer #1); Lennart Winkler (Reviewer #2).

Essential revisions:

1. A consistent critique from all reviewers concerned the need for improved clarity regarding experimental design, both in terms of the experimental steps themselves and also in how the experiments themselves explicitly relate back to the phenomena being studied. Numerous general and specific comments on this point are provided in the below reviews. Please also consider developing one or more figures to assist readers at a high level (while also not neglecting the need for major improvement in clarity and thoroughness at deeper levels in the text itself).

2. The above comments can be extended to the statistical frameworks applied in the paper, i.e. requiring more clarity, depth, and precision (and correction for multiple tests as appropriate in some cases). In some cases, this will likely require adjustment to the statistical approach used for a given hypothesis test.

3. Consider tempering slightly your conclusion that the effect of sexual conflict can be buffered by temperature in the wild, based on the experiments conducted to date.

*Reviewer #1 (Recommendations for the authors):*

Partly because the methods are placed at the end in this format, I had a lot of confusing moments going through the results. Certain details need to be explained before the methods section. For example, the fact that the polygamy treatment is also a highly male-biased treatment is very important and should be stated clearly. This will impact the interpretation of the results. It is also very important that you explain better the experimental design before diving into the results, otherwise much confusion follows. To be fair, even going through the methods section I found it quite hard to understand the details of the experimental design, especially the receptivity experiment part. It is a complicated experiment with a lot of aspects to it. It needs to be explained very carefully. Perhaps a diagram could help? In any case, you need to explain better the general architecture of the design before getting into the result section. Here are the facts that I think need to be mentioned before the result section so the results can be understood at all (either end of the introduction or the first paragraph of the Results section):

– There are several experiments in parallel, not just one, all across 3 temperatures.

– In experiment 1, flies are exposed to three temperatures in either monogamy (1:1) or polygamy (1f:3m) conditions. LRS is measured, senescence too, and behavior (courtship and aggression) on day 1 are recorded.

– In experiment 2, flies are also exposed to 3 temperatures. This part is very confusing in the methods l.508-525 and after reading it 5 times and drawing diagrams on my whiteboard I am still very unsure of what happened. As far as I got it, virgin individuals are collected and then paired repeatedly in monogamous assays. Males can either come from a "high sperm competition" pool (they are stored together with other virgin males) or "low competition" (males stored alone). There is also a duration factor which I understand to be how long the males sit in their respective competition and temperature treatments before the mating assays. What is clear is that this experiment leads to mating duration and latency measurements (the fact that this is not part of the "behavior" assays is also confusing while going through the results).

– Experiment 3, with the same male treatments as experiment 2, this time female fecundity and egg survival are measured. (where do we see the data from that experiment? As far as I could tell all the figures and supplementary figures are from experiments 1 and 2).

I apologize for the long review. I think it partly reflects my difficulty going through the results and methods section. Overall, I really liked the study even though I was frustrated at times by how much effort I had to put to understand certain figures or results. I think this can become a great manuscript if the methods and results are given just a bit more structure. I would also like to see some slight changes to the statistical methods and a couple of additional supplementary figures (female fecundity data and egg survival). See my details comments below.

Abstract

In the second mention of the mating system treatments, I would reiterate the term polyandry instead of using just "high male competition". Otherwise, it is not immediately clear that it is a mating system treatment. For example "At 20C, female senescence was accelerated under polyandry (high male competition)" and "At 28C, polyandry mostly resulted in reproductive ageing". As far as I understand these results come from Experiment 1 (following my numbering) so they do not correspond to the "male sperm competition" treatments of Experiments 2 and 3.

Not entirely clear results in the abstract. What does it mean when the authors say that at 28C female reproduction is "modulated"?

Results

This is a very dense result section with a lot of interesting data! I would suggest using post hoc tests to investigate the interaction effects, instead of running separate models per temperature. This way, you get p-values corrected for multiple testing. Regarding male-male aggression, I do not think it is correct to assign a zero for the monogamy treatments and test for interaction. The behavior is simply impossible in that treatment, and what you are really looking at is the main effect of temperature on the behavior in polyandry settings. Presenting it as you do is very confusing both in the text and Figure 4 and, I think, is not correct. To clarify the structure further, I would not use exactly the same colors for Polygamy/monogamy and high/low sperm competition. See detailed comments below. I also suggest clearly separating the results that come from different experiments so the readers know which sections are>

l.101: Maybe "temperature by mating system" instead of "x mating system" would be better when inline.

l.103: I would suggest running a post hoc test to see at which temperature the two mating systems differ or not, for example, Tukey's post hoc, instead of three separate models per temperature. What you do is also okay and I have no doubt that this is what the data is showing given how clear it is in Figure 1. But a post hoc would be nicer than 3 separate models because then you get proper p-values corrected for multiple testing.

You could use the package "emmeans" in R. See code below:

>library(emmeans)

>model<-lm(LRS~treatment*temperature, data=data)

>pairs(emmeans(model, ~temperature|treatment, adjust="Tukey"))

l.106: what is "H"?

l.107-109: I am confused about this part and the figure that goes with it. Where do those estimates come from? How are they calculated/simulated? See further comments below in the Figure section.

l.110: not exactly clear what model is run there. Is it only on time point 1 of Figure SI1? Please be more specific.

l.127-135: It is a bit difficult to interpret the model with its simplified estimate of reproductive senescence in the light of figure SI1 which shows the whole complexity of the data. It would be helpful to also have a figure with the mean (week 1, week 2) and mean (week 3, week 4). Do you justify why leaving out the last time point in the methods section?

l.130-131: It may be clearly significant, but it is not clear from the figure which way the effect is going. Perhaps if you provide a simplified figure with means it would help a bit. Also, same as above if you run post hoc tests you will have correct p-values and it should give you an estimate for the differences. The same comment goes for actuarial senescence.

l.147: aggression rate of…I am going to guess males towards females. Please specify. It could also be male-male (or female-female, at least in my study system…), but there can't be male-male aggression in a monogamy assay since there should be only one male. Same as above, please use post hoc tests to determine how the interaction is playing out.

l.147 again: I am super confused now, given that Figure 4a shows only 1 color (no monogamy values) and the figure legend says that we are looking at male-male aggression. Of course, male-male aggression should only be present in the polygamy treatment, but then what is the "estimated decrease"? is it compared to the zero of the monogamy treatment? And what is the interaction term in the model? Did you specify "zero male-male" aggression events in all the monogamy treatment observations? I don't think this is correct to use the monogamy treatment as a comparison here and to fit an interaction effect. What you are really looking at is the main effect of temperature on male-male aggression in polygamy treatments.

L.157: It would be interesting to also look at the rejection rate relative to the courtship rate. For example, you could see that in monogamy at 24C rejection rate seems to increase while the courtship rate decreases…probably not significant but still, another way to look at it that should be informative. Do females reject more, or do they just reject the same in proportion to courtship intensity?

l.162: I find it rather strange to title that section "ejaculate effects" since no ejaculates are sampled, weighed, or analyzed in any way. I understand that the assumption behind this setup is that there is male harm that can happen through ejaculate toxicity, but it is too much of an interpretation to bring that up here. What you measure is female mating behavior and female life history traits (fecundity, egg survival), the title of the section should reflect that.

l.165: Up to that point and before reading the methods section I was not even aware that there was a parallel experiment running with different treatments! That's not good. I got so lost the first time I encountered that part of the result section!! You need to (i) introduce better the experimental design and (ii) structure better the result section so we know where we are. Just the appearance of that "treatment duration" is confusing. If I understood correctly, it is the time that males are exposed to their respective temperature and competition treatments. If that is the case, why is the data not structured that way in Figure 5? We only see the different temperatures and "sperm competition" treatments, the duration of exposure is absent, which does not help.

L170: The same comment for 5b, why does the figure not represent the data, especially given the significant interaction between temperature and treatment duration?

l.179: I would like to see this data represented in a figure (supplementary is fine), including the treatment duration effect.

l.179: Just to emphasize again how it is absolutely needed to clarify the design and structure of the results: any reader who arrives at that point of the text and reads "for the number of eggs produced by females during the first 3 days…." Will assume that we are still talking about the same females as from the LRS data in the beginning and be utterly confused.

l.192: There is an effect of treatment duration on egg/offspring number and survival, but we do not even know in which direction. Please support with figures or give model estimates.

Discussion

I feel that the results from the different experiments remain too separated in the discussion. It is the opportunity to connect together the different pieces, and there are many in this complex puzzle. Is the behavior data consistent with the trend in the LRS data? The lifespan data? The fecundity data? There are some attempts in the last section but I feel that more can be done. The conclusion feels very generic.

l. 202: please remind the reader "net harm in terms of reduced LRS".

L204-207: after reading the methods, there are still no explanations about how those are calculated.

l.210: I would replace "Thus" with "More specifically".

l. 211: Did cold temperature increase female senescence? This is not what I would expect, and not what I see in Figure 3. Females live longer when it's cooler. The effect of the mating system is stronger at 20C though.

l. 212: modulated? In what way? Again, I think it is a bit of a leap to call that "ejaculate effects". Yes, it is likely part of the mechanism, but no you have not measured any ejaculate traits. I would be okay with a statement like "Male competition status affected female mating behavior and fecundity, likely through ejaculate trait. These effects were in turn affected by temperature".

l. 214: if the reproductive senescence is from the separate experiment on fecundity and egg survival, we still need to see the data somehow to be convinced of that statement.

l.236-241: this is actually the best explanation of the rate-sensitive estimates. It comes a bit late and still is not detailed enough.

l.303: ok but what about 28C? the male aggression is at the highest, courtship rate as high as 24C…yet no detectable male harm.

l.311-322: This info is super helpful and should come earlier to introduce this part of the experimental design.

l. 324: with respect to…

l.330-345: Very interesting. Is it possible that females are simply optimizing their mating rates by controlling the remating latency, instead of it depending entirely on the efficiency of SFPs? There are two sides to sexual conflict after all.

l.366: male harm, yes, but female behavior may also be plastic. For example, why see the female rejection rate as a consequence of the action of male SFP, rather than adaptive female behaviors? With this view, sexual selection is only between males, females are just part of the background… that's missing half of the picture.

l.371: I think you are over-simplifying here. Male harm is defined as a reduction in female LRS. At 20C there is a smaller decrease in LRS than at 24C, so reduced male harm, but a larger impact of polyandry on female survival. However, this reduction in survival need not be male harm, since it does not imply a reduction in fitness. It could be part of the female reproductive strategy.

Methods

l.483: I would remove the quotation marks on toxicity.

l.420: polyandry = biased sex ratio as well. Why? Monogamy vs polyandry could be 1:1 vs 3:3. The treatments are effectively "Monogamy" and "Polyandry with a highly male-biased sex ratio". This likely magnifies the effects of sexual conflict. If you were measuring male fitness, there would be additional problems such as underestimating selection on males. I think the sex-ratio unbalance needs to be mentioned more clearly in the first part of the manuscript. It is not an obscure detail to have only in the method section.

Figures

Figure 2: (a) female? Male reproductive success? Just from the result section and the figures, it is very difficult to understand what this is. Does it come from a simulation? Why do the values on the y-axis look so different from figure 1? If it is female LRS, why does it decrease with the population growth rate? So many questions and the few lines in the methods (l.572-577) do not answer them. Consider writing a supplementary page about how this works. (2b) the name of the y-axis is confusing. The cost is obviously highest at 24C but this is the lowest value on that axis so it is the opposite of a cost. Consider changing the variable to 1-X or changing the name. I would also avoid cutting the y-axis if possible (if you change to 1-X and still call it a cost, there is no need to show the zero so you can have your plot centered on the data without cutting the axis).

Figure 4: I understand the need to keep the figures compact but at first glance, the use of two different axes per figure is a bit of a headache. In the end, it would be alright not to have the "estimated decreases" in the main figures, because it is quite visible from the confidence intervals when the two treatments are different. But it is a matter of taste. Also, in Figure 4a it is a bit surprising to still see the estimated decrease when only one treatment (polyandry) is present. What is it compared to?

Figure 5: Keeping the same color schemes for the two experimental designs (polyandry/monogamy and high/low sperm competition) is confusing.

*Reviewer #2 (Recommendations for the authors):*

Thank you for the opportunity to read this exciting manuscript.

I found the methods detailed, yet I feel that at some points there is still crucial information missing that would be important to judge the robustness of results or for reproducibility (see detailed comments). Furthermore, while I applaud the authors for the great amount of data they collected and the experiments they combine here, I feel that the reader could benefit from a graphic experimental scheme or so to illustrate the procedures. I believe this might help to ease the digestion of the abundant amount of information necessarily given in the methods.

Detailed comments

– L90f: Could you provide more information regarding the source and year these temperature data were collected from?

– L105ff: Were the different harm levels significantly different from each other, as the abstract seems to suggest? I think here only the difference between monogamy and polygamy was tested within temperatures in a pair-wise manner, right?

– L107: 'Rate-sensitive fitness estimates…' Maybe add a bit more context here for an explanation of this term and why this was tested. (It has not been mentioned in the introduction, I think, so this comes a bit out of the blue.)

– Figure 1: I am confused about the model estimate at 20°C. The effect size is smaller compared to 28{degree sign}C and there seems to be just a difference in the CI between those two. This does not fit the data, I believe.

– Figure 2B: I think having a line at 1.0 for no relative fitness cost and then having a break in the y-axis is a bit misleading.

– L147: Male 'aggression rate'?

– L203: '…is not statistically detectable…' That is only a matter of statistical power, isn't it? Unless there is really zero difference, which seems unlikely given the presented data, I believe.

– Figure 4A: What is the comparison plotted here? 'Decrease' from where? In the other figures, it is the comparison between monogamy and polyandry, right?

– L330: '…this effect was consistent for males treated for 48h and 13 days…' Where is this shown? Wasn't there a 'sperm competition risk level x treatment duration interaction' (L164)?

– Figure 5: Are the data for the two experiments pooled here (i.e. 48h and 13 days)? Are these really comparable as the duration of exposure is so different and also the experimental protocol varies (i.e. emptied the seminal fluid and age difference between focal and competitors)?

– L373f: '…maintaining genetic variation in sexually selected traits in males…' I think this is true, but it has not been mentioned before and could do with a bit of context, I believe.

– L406: Could you specify how the temperature fluctuations were achieved? What type of device/incubator was used? Were these fluctuations random or in predictable/pre-programmed way?

– L413: What was the 'controlled density'?

– L415: Please specify, were all collected individuals in separate vials?

– L448: Please clearly define the 'W' in this formula.

– L460f: I am slightly confused by this sentence: 'However, all flies were 5 days old at the start of the experiment.' Do you mean the experiment in general? Then I am not sure what this sentence is here for. Or, do you mean by the start of each behavioral essay? Then I don't understand how the order could have been randomized. Please clarify.

– L522: What exactly does 'right-censored' mean for your analysis?

– L537: How exactly was this modelled? I am not sure that this is sufficient information to be repeatable.

– L550: Maybe add a citation for the R packages used in the analysis?

– Unfortunately, I was not able to access the data or the code via the provided link. Please make sure these are working. I greatly appreciate the publication of code and data.*Reviewer #3 (Recommendations for the authors):*

I have a number of recommendations that authors should address before the paper is suitable for publication (in the order in which they arise in the manuscript).

1) The authors should avoid detailing the relative harm index H in the abstract. This index will have no meaning to most readers and requires them to read the details of the methods to understand, which is contrary to the purpose of the abstract.

2) L101: Here and throughout the paper it is unclear what statistical models are being tested and how. For example, here the authors test for a significant interaction between temperature and mating system on LRS, and report a chi-square statistic. In the methods, they state that they test the "compared GLMs with their corresponding null GLMs using likelihood ratio test" (L559), so I am guessing that this is the outcome of the test. What is the null model, however? Is it the model without the interaction and just the main effects? They are not explicit here. Further, tests of the main effects are generally considered unimportant when adjusting for the interaction (although trends in the main effects can nevertheless be interpreted). Did they adjust for the interaction when testing for the main effects? Again, this is unclear from the statistical details. I did not have access to the R script (which is stated as being available on Dryad but appears not to be), so am unsure as to what the tests actually are. It may be more straightforward to report the results of the GLMs with an ANOVA table.

3) The authors subsequently test the effect of the mating system on LRS at each temperature and report the harm index. Since these are posthoc tests, they should apply a Bonferoni correction to the significance levels (which may render the effect of the mating system on LRS at 20°C non-significant.

4) Figure 1. Here and in many of the other figures, the authors report both the means (left y-axis) and the contrasts (right y-axis), the latter being labelled 'estimated decrease in LRS'. These estimated decreases look like they are the parameter estimates from the GLMs, but their value depends on how the data are coded. Here they are the decrease in LRS in the polyandrous versus the monogamous, but they could equally be the increase in LRS in the monogamous versus the polyandrous if the authors had coded their data differently. Thus simply labeling the axis 'estimated decrease in LRS' is not sufficient. I think that these parameter values would be best presented in a table, rather than included in the figure. It would be also preferable if the authors included all the data in the figures, for example using a violin plot, rather than just the means and the 95% Cis, since this provides the reader additional information about the distribution of the data.

5) Figure 2. This is perhaps a lack of familiarity on my part, but I do not know how these figures were generated or what they show. These are presumably the outputs of a model (there are some details provided at L575-577) but much more detail needs to be given here.

6) L136. I appreciate that the authors want to provide the details of the statistical analyses on female survival, but all the parentheses make this paragraph extremely difficult to follow. Perhaps including the results of the analysis as a table would make this much easier to read.

7) L146. Again, this paragraph is complex and difficult to follow. The authors may want to state the main finding of their data, before detailing the statistics that support this finding. As for the analysis of the LRS, the P-values of the main effects are questionable if they have been adjusted for significant interaction (which is what is implied in the use of type III ANOVA, L560). As an alternative the authors could conduct posthoc tests on the effect of temperature or mating system for each mating system or temperature respectively, applying appropriate corrections for a multiplicity of tests, e.g. Bonferroni. As for Figure 1, the parameter values for the main effects in Figure 4 (right y-axis) would be best presented in a supplementary table.

8) I find the evidence that temperature affects ejaculate quality to harm females much less convincing. The authors report data on the effects of the mating system on mating duration and remating latency in females mating with males that have been kept singularly (low sperm competition) or in groups of three (high sperm competition). They should clearly explain the relationship between sperm competition, ejaculate quality, and these two assays, with citations if these methods have been used before. The description of these experiments is particularly difficult to follow, but it appears that they recorded mating duration both for the first mating (with males having just been exposed to different levels of perceived sperm competition) and during rematings. Which of these is shown in Figure 5A? They also conducted mating assays on males that had been maintained at different temperatures for 48h or 13 days (L 502). Data from which of these are used in Figure 5A? Which of these were used for female remating latency? Because the experimental methods are so difficult to follow, it precludes an interpretation of the data and makes it difficult to determine whether the data support the conclusions of the authors.

9) L179: This is another dense paragraph that essentially shows that sperm competition risk does not affect female fecundity and survival. That is, while sperm competition may increase mating duration, or remating latency, these do not appear to result in female harm. This undermines the interpretation that changes in mating duration and remating latency with risk of sperm competition reflect the chemistry of the ejaculate. An alternative explanation is that maintaining males with other males may increase his mating duration, which in turn leads to an increase in remating latency. It is interesting to note that the interaction between temperature and risk of sperm competition on mating duration and remating latency reflects a decline in these dependent factors with temperature when risk is low, but maintenance when risk is high. Chemical analysis of the ejaculate would help clarify the relationship between temperature and sperm competition on post-copulatory female harm.

---

## [Author Response]

Reviewer #1 (Recommendations for the authors):Partly because the methods are placed at the end in this format, I had a lot of confusing moments going through the results. Certain details need to be explained before the methods section. For example, the fact that the polygamy treatment is also a highly male-biased treatment is very important and should be stated clearly. This will impact the interpretation of the results. It is also very important that you explain better the experimental design before diving into the results, otherwise much confusion follows. To be fair, even going through the methods section I found it quite hard to understand the details of the experimental design, especially the receptivity experiment part. It is a complicated experiment with a lot of aspects to it. It needs to be explained very carefully. Perhaps a diagram could help? In any case, you need to explain better the general architecture of the design before getting into the result section. Here are the facts that I think need to be mentioned before the result section so the results can be understood at all (either end of the introduction or the first paragraph of the Results section):

Excellent points/suggestions. See responses below.

- There are several experiments in parallel, not just one, all across 3 temperatures.- In experiment 1, flies are exposed to three temperatures in either monogamy (1:1) or polygamy (1f:3m) conditions. LRS is measured, senescence too, and behavior (courtship and aggression) on day 1 are recorded.- In experiment 2, flies are also exposed to 3 temperatures. This part is very confusing in the methods l.508-525 and after reading it 5 times and drawing diagrams on my whiteboard I am still very unsure of what happened. As far as I got it, virgin individuals are collected and then paired repeatedly in monogamous assays. Males can either come from a "high sperm competition" pool (they are stored together with other virgin males) or "low competition" (males stored alone). There is also a duration factor which I understand to be how long the males sit in their respective competition and temperature treatments before the mating assays. What is clear is that this experiment leads to mating duration and latency measurements (the fact that this is not part of the "behavior" assays is also confusing while going through the results).- Experiment 3, with the same male treatments as experiment 2, this time female fecundity and egg survival are measured. (where do we see the data from that experiment? As far as I could tell all the figures and supplementary figures are from experiments 1 and 2).

We agree with the referee that all the facts pointed out need to be clarified before the Results section. To do so, we have reworded and reorganized both the methods and results. More specifically, we have moved the methods to precede the results, and added diagrams for all experiments (Figures S1.1 to S1.5) and a general figure (Figure 1) that reflects the overall design. We now also describe the methods and results independently for each experiment. Additionally, we now provide additional figures in the supplementary material that include the egg-to-adult viability data.

I apologize for the long review. I think it partly reflects my difficulty going through the results and methods section. Overall, I really liked the study even though I was frustrated at times by how much effort I had to put to understand certain figures or results. I think this can become a great manuscript if the methods and results are given just a bit more structure. I would also like to see some slight changes to the statistical methods and a couple of additional supplementary figures (female fecundity data and egg survival). See my details comments below.

Again, thanks for the effort. We agree with all these suggestions. Details follow on how we have dealt with them.

AbstractIn the second mention of the mating system treatments, I would reiterate the term polyandry instead of using just "high male competition". Otherwise, it is not immediately clear that it is a mating system treatment. For example "At 20C, female senescence was accelerated under polyandry (high male competition)" and "At 28C, polyandry mostly resulted in reproductive ageing". As far as I understand these results come from Experiment 1 (following my numbering) so they do not correspond to the "male sperm competition" treatments of Experiments 2 and 3.Not entirely clear results in the abstract. What does it mean when the authors say that at 28C female reproduction is "modulated"?

We now use the term polyandry to explain our results in the abstract and also re-phrased the sentences to make them clearer. “At 20°C, male harassment of females was reduced, and polyandry accelerated female actuarial ageing. In contrast, the effect of mating on female receptivity (a component of ejaculate toxicity) was affected at 28ºC, where the mating costs for females decreased and polyandry mostly resulted in accelerated reproductive ageing”.

What we meant with “at 28°C female reproduction is modulated” is that, as we now describe in the abstract, female receptivity was only affected by male sperm competition risk at 28°C.

ResultsThis is a very dense result section with a lot of interesting data! I would suggest using post hoc tests to investigate the interaction effects, instead of running separate models per temperature. This way, you get p-values corrected for multiple testing.

As we now explain in the manuscript (lines 314-324), we did not use Tukey contrasts for our original models because we modelled temperature as a continuous variable (which it is). However, we now use the Benjamini-Hochberg method to control for multiple testing. To err on the side of caution, we also re-run our analyses using temperature as a categorical factor, which allowed us to run Tukey contrasts as an additional way to explore interactions while correcting for multiple testing. Both approaches yield qualitatively identical results and we present the latter models in the SM (tables S1.1, S1.2c, S3.1 and S4.2).

Regarding male-male aggression, I do not think it is correct to assign a zero for the monogamy treatments and test for interaction. The behavior is simply impossible in that treatment, and what you are really looking at is the main effect of temperature on the behavior in polyandry settings. Presenting it as you do is very confusing both in the text and Figure 4 and, I think, is not correct.

The reviewer is right. We re-run the model looking at male-male aggression as suggested and report these results in the current version (lines 415-416). “For male-male aggression rate we detected a significant temperature effect (F1,214 = 14.45; p = 0.0003 Figure 5c and S5.1c)”.

To clarify the structure further, I would not use exactly the same colors for Polygamy/monogamy and high/low sperm competition. See detailed comments below. I also suggest clearly separating the results that come from different experiments so the readers know which sections are:

We agree with this and have used different colours for polyandry/monogamy and high/low sperm competition risk experiments on the respective figures. More specifically, on figures 2 to 5 (and Figures S2.1-2, S4.1 and S5.1-2) we used orange and violet to differentiate between polyandry and monogamy, respectively, and on figure 6 (and Figures S6.1-4) we used yellow and blue to differentiate between high vs low sperm competition risk. We have also separated the results by experiments.

l.101: Maybe "temperature by mating system" instead of "x mating system" would be better when inline.

Done. Throughout all the descriptions of the results we have used “by” instead of “x” to report the interactions between effects.

l.103: I would suggest running a post hoc test to see at which temperature the two mating systems differ or not, for example, Tukey's post hoc, instead of three separate models per temperature. What you do is also okay and I have no doubt that this is what the data is showing given how clear it is in Figure 1. But a post hoc would be nicer than 3 separate models because then you get proper p-values corrected for multiple testing.You could use the package "emmeans" in R. See code below:>library(emmeans)>model<-lm(LRS~treatment*temperature, data=data)>pairs(emmeans(model, ~temperature|treatment, adjust="Tukey"))

Thank you for the suggestion and the code. As we explained before, we originally did not use Tukey contrasts for our original models because we modelled temperature as a continuous variable. However, we also re-run our analyses using temperature as a categorical factor, which allowed us to run Tukey contrasts. We present those results in the SM (see tables S1.1, S1.2c, S3.1 and S4.2).

l.106: what is "H"?

Clarified in the methods section (lines 177-181) before it appears in the results and discussion. “…by calculating the relative harm (H) following Yun et al. (2021):…”

l.107-109: I am confused about this part and the figure that goes with it. Where do those estimates come from? How are they calculated/simulated? See further comments below in the Figure section.

In the current version (lines 188-201), we detail how we calculate rate-sensitive fitness estimates. Briefly, based on our LRS measures we calculated both individual (ω_ind_) and population (ω_pop_) ratesensitive estimates for different intrinsic rates of population growth (r) following the procedure detailed in Edward et al. 2011. We also calculated the relative cost of polyandry for each temperature treatment across different values of r, using the population rate-sensitive estimate.

l.110: not exactly clear what model is run there. Is it only on time point 1 of Figure SI1? Please be more specific.

We partitioned overall LRS effects into effects on early reproductive rate (i.e., offspring produced during the first two weeks of age), actuarial ageing (i.e., lifespan), and reproductive ageing (i.e., offspring produced over weeks 1-2 vs. 3-4) that we analysed separately fitting generalized linear models. We also re-made the figure (Figure S2.2), which now shows the mean of each component. This is explained in lines 182-184 and 325-329.

l.127-135: It is a bit difficult to interpret the model with its simplified estimate of reproductive senescence in the light of figure SI1 which shows the whole complexity of the data. It would be helpful to also have a figure with the mean (week 1, week 2) and mean (week 3, week 4). Do you justify why leaving out the last time point in the methods section?

Agree. We have re-made figure SI1 (now Figure S2.2) with the changes suggested by the reviewer. See lines 185-187 in the manuscript for justification about not using week 5 to calculate late reproductive rate and reproductive ageing. “We used weeks 3-4 as an estimate of late reproductive rate because mortality was already evident at this point (i.e., reflecting ageing) and then was very high from week 5 onwards (Figure S4.1; thus preventing accurate estimation of reproductive success)”.

l.130-131: It may be clearly significant, but it is not clear from the figure which way the effect is going. Perhaps if you provide a simplified figure with means it would help a bit. Also, same as above if you run post hoc tests you will have correct p-values and it should give you an estimate for the differences. The same comment goes for actuarial senescence.

As suggested, we have implemented that change (Figure S2.2) and provided the estimates for the differences (tables 1 and S1.1a).

l.147: aggression rate of…I am going to guess males towards females. Please specify. It could also be male-male (or female-female, at least in my study system…), but there can't be male-male aggression in a monogamy assay since there should be only one male. Same as above, please use post hoc tests to determine how the interaction is playing out.

Totally agree. We clarify this in the text and re-run the model looking at male-male aggression only for the polyandry mating system (lines 350-352). We report these results in the current version (lines 415-416).

l.147 again: I am super confused now, given that Figure 4a shows only 1 color (no monogamy values) and the figure legend says that we are looking at male-male aggression. Of course, male-male aggression should only be present in the polygamy treatment, but then what is the "estimated decrease"? is it compared to the zero of the monogamy treatment? And what is the interaction term in the model? Did you specify "zero male-male" aggression events in all the monogamy treatment observations? I don't think this is correct to use the monogamy treatment as a comparison here and to fit an interaction effect. What you are really looking at is the main effect of temperature on male-male aggression in polygamy treatments.

We apologize for the misunderstanding. As explained in the previous comment, we have now run the model checking for male-male aggression only for the polyandry mating system (lines 350-352). Now the figure does not show any estimated decrease (Figure 5c).

L.157: It would be interesting to also look at the rejection rate relative to the courtship rate. For example, you could see that in monogamy at 24C rejection rate seems to increase while the courtship rate decreases…probably not significant but still, another way to look at it that should be informative. Do females reject more, or do they just reject the same in proportion to courtship intensity?

Thank you for the suggestion. We have implemented an analysis to explore potential differences in female rejections per courtship (see results in lines 413-415). “We did not detect a significant interaction between temperature and mating system (F1,294 = 0.30; p = 0.582), nor a temperature effect (F1,294 = 0.08; p = 0.773), for rejection rates per courtship.”

l.162: I find it rather strange to title that section "ejaculate effects" since no ejaculates are sampled, weighed, or analyzed in any way. I understand that the assumption behind this setup is that there is male harm that can happen through ejaculate toxicity, but it is too much of an interpretation to bring that up here. What you measure is female mating behavior and female life history traits (fecundity, egg survival), the title of the section should reflect that.

We have changed “ejaculate effects” by “mating effects”.

l.165: Up to that point and before reading the methods section I was not even aware that there was a parallel experiment running with different treatments! That's not good. I got so lost the first time I encountered that part of the result section!! You need to (i) introduce better the experimental design and (ii) structure better the result section so we know where we are. Just the appearance of that "treatment duration" is confusing. If I understood correctly, it is the time that males are exposed to their respective temperature and competition treatments. If that is the case, why is the data not structured that way in Figure 5? We only see the different temperatures and "sperm competition" treatments, the duration of exposure is absent, which does not help.

For better understanding, as we explained in previous comments, the methods section is now placed before the results. We re-structured both sections, and extensively revised the content to address all the issues that were raised by the reviewer. We also added diagrams for all experiments (Figures S1.1 to S1.5) and an overall figure that outlines the design of the study (Figure 1). We also re-made figure 5 (now is the figure 6) to include both treatment durations. We also supply a table with the estimates for the differences (table 3).

L170: The same comment for 5b, why does the figure not represent the data, especially given the significant interaction between temperature and treatment duration?

Initially, we combined the data from two experiments, which lasted 48 hours and 13 days, solely to demonstrate the general impact of sperm competition risk, as it remained consistent across both treatment durations. However, in the present version of our manuscript, we have created a new figure 6 (formerly figure 5), which shows the results from each experiment separately.

l.179: I would like to see this data represented in a figure (supplementary is fine), including the treatment duration effect.

Done. Please, see figures S6.2 and S6.3.

l.179: Just to emphasize again how it is absolutely needed to clarify the design and structure of the results: any reader who arrives at that point of the text and reads "for the number of eggs produced by females during the first 3 days…." Will assume that we are still talking about the same females as from the LRS data in the beginning and be utterly confused.

Again, thank you for the suggestion. We have reorganized the methods and Results sections and made significant revisions to the content in order to address the aforementioned point. We have separated the methods and results by experiments and directly pointed them out at the beginning of each section.

l.192: There is an effect of treatment duration on egg/offspring number and survival, but we do not even know in which direction. Please support with figures or give model estimates.

Done. Please, see figures S6.2 and S6.3 and tables 4 and S4.1.

DiscussionI feel that the results from the different experiments remain too separated in the discussion. It is the opportunity to connect together the different pieces, and there are many in this complex puzzle. Is the behavior data consistent with the trend in the LRS data? The lifespan data? The fecundity data? There are some attempts in the last section but I feel that more can be done. The conclusion feels very generic.

We have re-written some parts of the discussion to better integrate all the different results. We divided the discussion in two main sections: “Temperature effects on male harm and its consequences for populations” and “Temperature effects on sexual conflict mechanisms in *Drosophila melanogaster*”. Across the latter, we discuss whether the behaviour (lines 586-596) and fecundity (lines 607-610, 626-631) data is consistent with the trend in the LRS data.

l. 202: please remind the reader "net harm in terms of reduced LRS".

Done. “…net harm to females varies markedly within this range, such that relative harm (i.e., the proportional reduction in average female LRS in polygamy vs. monogamy) was…” (lines 471-473).

L204-207: after reading the methods, there are still no explanations about how those are calculated.

In the current version (lines 188-201), we detail how we calculate rate-sensitive fitness estimates.

l.210: I would replace "Thus" with "More specifically".

Done.

l. 211: Did cold temperature increase female senescence? This is not what I would expect, and not what I see in Figure 3. Females live longer when it's cooler. The effect of the mating system is stronger at 20C though.

At cold temperatures polyandry accelerated female actuarial ageing compared with warm temperatures. We re-phrased the paragraph accordingly (lines 480-482).

l. 212: modulated? In what way? Again, I think it is a bit of a leap to call that "ejaculate effects". Yes, it is likely part of the mechanism, but no you have not measured any ejaculate traits. I would be okay with a statement like "Male competition status affected female mating behavior and fecundity, likely through ejaculate trait. These effects were in turn affected by temperature".

We hope this is clear in the current version: “Warm temperatures impacted mating costs and effects on female receptivity (i.e., post-copulatory mechanism), and female fitness decreased more via accelerated reproductive ageing than at colder temperatures” (lines 482-484).

l. 214: if the reproductive senescence is from the separate experiment on fecundity and egg survival, we still need to see the data somehow to be convinced of that statement.

We now present the reproductive senescence data in figure S2.2.

l.236-241: this is actually the best explanation of the rate-sensitive estimates. It comes a bit late and still is not detailed enough.

As suggested, we have gone through the explanation in a more detailed way (lines 188-201).

l.303: ok but what about 28C? the male aggression is at the highest, courtship rate as high as 24C…yet no detectable male harm.

We clarified in lines 593-596 that the decrease in male harm that we detected at 20° vs 24 ° is likely explained by the substantial drop in male harassment at this temperature. However, the same logic cannot apply for the decrease in male harm to females that we detected at 28°C vs. 24°C. Instead, receipt of a male ejaculate resulted in a sharper decrease of female receptivity in high vs. low SCR at 28°C (lines 627-631).

l.311-322: This info is super helpful and should come earlier to introduce this part of the experimental design.

Now it is part of the methodology, and thus comes before explaining the experimental design (lines 239-247).

l. 324: with respect to…

Changed.

l.330-345: Very interesting. Is it possible that females are simply optimizing their mating rates by controlling the remating latency, instead of it depending entirely on the efficiency of SFPs? There are two sides to sexual conflict after all.

Agree. We have extended this idea (lines 636-647).

l.366: male harm, yes, but female behavior may also be plastic. For example, why see the female rejection rate as a consequence of the action of male SFP, rather than adaptive female behaviors? With this view, sexual selection is only between males, females are just part of the background… that's missing half of the picture.

We agree with the fact that temperature may affect female behaviour or physiology in a way that makes them more resistant to harm. We have included the female perspective in the current discussion (lines 636-653).

l.371: I think you are over-simplifying here. Male harm is defined as a reduction in female LRS. At 20C there is a smaller decrease in LRS than at 24C, so reduced male harm, but a larger impact of polyandry on female survival. However, this reduction in survival need not be male harm, since it does not imply a reduction in fitness. It could be part of the female reproductive strategy.

We have discussed the point that the reviewer highlights throughout the last part of the Discussion section (lines 636-647).

Methodsl.483: I would remove the quotation marks on toxicity.

Done.

l.420: polyandry = biased sex ratio as well. Why? Monogamy vs polyandry could be 1:1 vs 3:3. The treatments are effectively "Monogamy" and "Polyandry with a highly male-biased sex ratio". This likely magnifies the effects of sexual conflict. If you were measuring male fitness, there would be additional problems such as underestimating selection on males. I think the sex-ratio unbalance needs to be mentioned more clearly in the first part of the manuscript. It is not an obscure detail to have only in the method section.

We clarify now in the manuscript that the sex ratio in this species is typically 1:1, however the operational sex ratio is male-biased and frequently reaches a 3:1 (or higher) male-bias in mating patches in the wild. Thus, the 1:1 vs. 3:1 sex ratios used in this study represent biologically relevant scenarios and have actually become standard in *Drosophila* studies measuring male harm to females. See lines 150-155.

FiguresFigure 2: (a) female? Male reproductive success? Just from the result section and the figures, it is very difficult to understand what this is. Does it come from a simulation? Why do the values on the y-axis look so different from figure 1? If it is female LRS, why does it decrease with the population growth rate? So many questions and the few lines in the methods (l.572-577) do not answer them. Consider writing a supplementary page about how this works. (2b) the name of the y-axis is confusing. The cost is obviously highest at 24C but this is the lowest value on that axis so it is the opposite of a cost. Consider changing the variable to 1-X or changing the name. I would also avoid cutting the y-axis if possible (if you change to 1-X and still call it a cost, there is no need to show the zero so you can have your plot centered on the data without cutting the axis).

In the current version, we detail how we calculate rate-sensitive fitness estimates (lines 188-201). We also implemented changes in figure 2 (now figure 3). Panel 3a shows the average fitness estimate of individual females (W_ind_) for different population growth rates. We have implemented the change suggested by the reviewer for the panel 3b.

Figure 4: I understand the need to keep the figures compact but at first glance, the use of two different axes per figure is a bit of a headache. In the end, it would be alright not to have the "estimated decreases" in the main figures, because it is quite visible from the confidence intervals when the two treatments are different. But it is a matter of taste. Also, in Figure 4a it is a bit surprising to still see the estimated decrease when only one treatment (polyandry) is present. What is it compared to?

We have left the figures only with the means and now provide tables with the estimates to explore size effects.

Figure 5: Keeping the same color schemes for the two experimental designs (polyandry/monogamy and high/low sperm competition) is confusing.

We have changed the colour for the experiments 2 to 5 (high/low sperm competition risk). Please, see this change in figures 6 and S6.1-4.

Reviewer #2 (Recommendations for the authors):Thank you for the opportunity to read this exciting manuscript.I found the methods detailed, yet I feel that at some points there is still crucial information missing that would be important to judge the robustness of results or for reproducibility (see detailed comments). Furthermore, while I applaud the authors for the great amount of data they collected and the experiments they combine here, I feel that the reader could benefit from a graphic experimental scheme or so to illustrate the procedures. I believe this might help to ease the digestion of the abundant amount of information necessarily given in the methods.

We completely agree. Please notice that to clarify the methods we have placed this section before the Results section and re-written both the methods and results. This has significantly improved the clarity of the manuscript. We have also included several diagrams to illustrate all our experiments (in the SM. Figures S1.1-5) along with a general schematic figure of the whole design that we present early on (in the introduction, see Figure 1).

Detailed comments– L90f: Could you provide more information regarding the source and year these temperature data were collected from?

Sure. The data come from WorldClim2, which has average monthly climate data for minimum, mean, and maximum temperature and for precipitation for 1970-2000 (Fick and Hijmans, 2017). Now, it is referenced in line 92.

– L105ff: Were the different harm levels significantly different from each other, as the abstract seems to suggest? I think here only the difference between monogamy and polygamy was tested within temperatures in a pair-wise manner, right?

As we detected a significant interaction between temperature by mating system for LRS, we interpreted that net male harm varies markedly among temperatures. To explore the nature of the interaction we ran models separately for each temperature in which only the difference between monogamy and polyandry is tested. We found a larger effect of mating system on LRS at 24°C than at 20°C, and no effect at 28°C (tables 1 and S1.1a).

– L107: 'Rate-sensitive fitness estimates…' Maybe add a bit more context here for an explanation of this term and why this was tested. (It has not been mentioned in the introduction, I think, so this comes a bit out of the blue.)

In the current version, we detail how we calculated rate-sensitive fitness estimates (lines 188-201).

– Figure 1: I am confused about the model estimate at 20°C. The effect size is smaller compared to 28{degree sign}C and there seems to be just a difference in the CI between those two. This does not fit the data, I believe.

We have double-checked the data and re-run the analysis to be sure about the output of the models. Separate models for each temperature level show an estimate at 20°C = 1.07, 95% CI 0.06 -2.06 and at 28°C = 1.88, 95% CI -0.58 – 4.36 (table 1). However, note that the Tukey’s post hoc test yielded the same results for mating system effect as running the models separately for each temperature, albeit with a greater estimate at 20° than at 28° (table S1.1a-LRS-).

– Figure 2B: I think having a line at 1.0 for no relative fitness cost and then having a break in the y-axis is a bit misleading.

We agree with this. We have re-made figure 2b (now figure 3b) and now the population costs are shown as 1 – Cr for better understanding.

– L147: Male 'aggression rate'?

We meant male aggression rate. Clarified in line 415.

– L203: '…is not statistically detectable…' That is only a matter of statistical power, isn't it? Unless there is really zero difference, which seems unlikely given the presented data, I believe.

We have re-formulated the phrase to avoid misunderstandings (lines 471-473). “…that relative harm (i.e., the proportional reduction in average female LRS in polygamy vs. monogamy) was maximal at 24ºC, decreased at 20ºC and was minimum at 28°C…”.

– Figure 4A: What is the comparison plotted here? 'Decrease' from where? In the other figures, it is the comparison between monogamy and polyandry, right?

We have re-run the model looking at male-male aggression only for the polyandry mating system. We report these results in the current version (lines 415-416). We re-made figure 4 (now figure 5) and you can see that now there is not a comparison plotted for male-male aggression rate (Figure 5c), but for other figures the comparison is between monogamy and polyandry (Figure 5a-b).

– L330: '…this effect was consistent for males treated for 48h and 13 days…' Where is this shown? Wasn't there a 'sperm competition risk level x treatment duration interaction' (L164)?

We re-made figure 5 (now figure 6), and now show both treatment durations in the figure. We also supply a table with the estimates for the differences (Table 3).

– Figure 5: Are the data for the two experiments pooled here (i.e. 48h and 13 days)? Are these really comparable as the duration of exposure is so different and also the experimental protocol varies (i.e. emptied the seminal fluid and age difference between focal and competitors)?

Yes, the data for the two experiments (48h and 13 days) were pooled initially because we wanted to focus on the overall effect of the sperm competition risk, as it was consistent across treatment durations. However, in the current version of the manuscript, we re-made figure 5 (now figure 6) and show both experiments independently.

– L373f: '…maintaining genetic variation in sexually selected traits in males…' I think this is true, but it has not been mentioned before and could do with a bit of context, I believe.

Now this idea is also considered at the first part of the Discussion section (lines 537-546). As a potential consequence arising from the observed temperature effects on male harm mechanisms, we propose that environmental variability may foster the maintenance of genetic variation underlying different mechanisms of male harm, and even potentially divergent male-male competition strategies.

– L406: Could you specify how the temperature fluctuations were achieved? What type of device/incubator was used? Were these fluctuations random or in predictable/pre-programmed way?

We pre-programmed daily temperature fluctuations (24 ±4°C) mimicking natural daily temperature conditions during the reproductively active season using a Pol Eko ST 1200 incubator. We now clarify this in lines 124-126.

– L413: What was the 'controlled density'?

Every time we collected eggs to obtain experimental flies, we used a controlled density of ~200eggs (223 ± 14.3, mean and 95% CI), following the procedure described by Clancy and Kennington, 2001. We now specify this in line 137.

– L415: Please specify, were all collected individuals in separate vials?

Once we collected and sexed virgin experimental flies, the way that we kept them until their use depended on the experiment. For example, for the experiment 1 we collected virgin flies into same-sex vials of 15 individuals (line 156). However, for the experiments 2 to 5 we placed the experimental males in vials either individually (low sperm competition risk) or in a same-sex group of 8 (high sperm competition risk; clarified in lines 264-266) before we allocated them to the different temperature treatments.

– L448: Please clearly define the 'W' in this formula.

To calculate relative harm (H), we followed the formula described by Yun et al. (2021) where W corresponds to female’s fitness. We now clarify this in lines 177-181.

– L460f: I am slightly confused by this sentence: 'However, all flies were 5 days old at the start of the experiment.' Do you mean the experiment in general? Then I am not sure what this sentence is here for. Or, do you mean by the start of each behavioral essay? Then I don't understand how the order could have been randomized. Please clarify.

Indeed, we have worked in the current version to make the method section easier to follow and more comprehensive (lines 202-210). What we meant is that, for experiment 1, we conducted behavioural observations in the same temperature control room, due to logistic limitations. So, we had to conduct trials at 20, 24 and 28°C over three consecutive days (because the same CT room can only be kept at a given temperature at the same time). However, as we collected virgin flies over three consecutive days, we ensured all flies were 5 days-old at the start of the experiment (i.e., when the observations were done at each temperature). We conducted observations first at 20°, then at 28° and finally at 24°C (we haphazardly chose this order).

– L522: What exactly does 'right-censored' mean for your analysis?

It means individuals that are taken into account for demographic analysis until the day they disappear (Kleinbaum and Klein, 2012). We explain these terms in lines 345-346.

– L537: How exactly was this modelled? I am not sure that this is sufficient information to be repeatable.

We have clarified how the fecundity and survival assays were done (lines 293-310).

– L550: Maybe add a citation for the R packages used in the analysis?

Done (lines 343-344).

– Unfortunately, I was not able to access the data or the code via the provided link. Please make sure these are working. I greatly appreciate the publication of code and data.

We apologize for this. We provide the code and data directly just in case the link from Dryad doesn’t work yet.

Reviewer #3 (Recommendations for the authors):I have a number of recommendations that authors should address before the paper is suitable for publication (in the order in which they arise in the manuscript).1) The authors should avoid detailing the relative harm index H in the abstract. This index will have no meaning to most readers and requires them to read the details of the methods to understand, which is contrary to the purpose of the abstract.

Agree, we now express this in %.

2) L101: Here and throughout the paper it is unclear what statistical models are being tested and how. For example, here the authors test for a significant interaction between temperature and mating system on LRS, and report a chi-square statistic. In the methods, they state that they test the "compared GLMs with their corresponding null GLMs using likelihood ratio test" (L559), so I am guessing that this is the outcome of the test. What is the null model, however? Is it the model without the interaction and just the main effects? They are not explicit here. Further, tests of the main effects are generally considered unimportant when adjusting for the interaction (although trends in the main effects can nevertheless be interpreted). Did they adjust for the interaction when testing for the main effects? Again, this is unclear from the statistical details. I did not have access to the R script (which is stated as being available on Dryad but appears not to be), so am unsure as to what the tests actually are. It may be more straightforward to report the results of the GLMs with an ANOVA table.

We have amended the statistical analyses part of the manuscript. Briefly, we have used ANOVA type III (via “F” statistic) to compute p-values corrected by the Benjamini-Hochberg method. We also compared GLMs with their corresponding null GLMs in order to test the significance of the independent variables in the full model (omnibus test). As we used ANOVA type III, it is true that the test of the main effects could be considered unimportant. However, we feel interpretation of interactions and main effects is straightforward given the raw data shown in the figures. In addition, we re-run all analysis and conducted Tukey’s post hoc tests on the effect of mating system for each temperature (as well as on the effect of treatment duration for each mating system; see Tables S1-3), when significant interactions were detected. As you will see, results are very consistent with our initial interpretation of interaction and main effects. We are sorry the reviewer could not get access to the R script. We have contacted Dryad but also provide the code and data in the submission, in case the Dryad link hasn’t been amended by the time the reviewer sees this.

3) The authors subsequently test the effect of the mating system on LRS at each temperature and report the harm index. Since these are posthoc tests, they should apply a Bonferoni correction to the significance levels (which may render the effect of the mating system on LRS at 20°C non-significant.

We thank the reviewer for this suggestion. As we now explain in the manuscript (lines 328-338) we repeated all analyses and applied the Benjamini-Hochberg method to control for multiple testing (the best option given we fit temperature as a continuous variable). Furthermore, we also repeated our analyses fitting temperature as a categorical factor and using Tukey contrasts to further examine interactions while accounting for multiple testing. Both methods produced very similar results, and we included the latter approach in the supplementary material (tables S1.1, S1.2c, S3.1 and S4.2).

4) Figure 1. Here and in many of the other figures, the authors report both the means (left y-axis) and the contrasts (right y-axis), the latter being labelled 'estimated decrease in LRS'. These estimated decreases look like they are the parameter estimates from the GLMs, but their value depends on how the data are coded. Here they are the decrease in LRS in the polyandrous versus the monogamous, but they could equally be the increase in LRS in the monogamous versus the polyandrous if the authors had coded their data differently. Thus simply labeling the axis 'estimated decrease in LRS' is not sufficient. I think that these parameter values would be best presented in a table, rather than included in the figure. It would be also preferable if the authors included all the data in the figures, for example using a violin plot, rather than just the means and the 95% Cis, since this provides the reader additional information about the distribution of the data.

Attending the suggestions by two of the reviewers, we have left the figures only with the means and now we provide tables with the estimates. We preferred to leave the main figures showing the mean +/- SEM, considering that in this way differences are easier to interpret. However, we supply violin plots in the SM (Figures S2.1-2, S4.1, S5.1-2 and S6.1-4).

5) Figure 2. This is perhaps a lack of familiarity on my part, but I do not know how these figures were generated or what they show. These are presumably the outputs of a model (there are some details provided at L575-577) but much more detail needs to be given here.

We agree with the lack of clarity about how figure 2 (now figure 3) was generated. In the current version, you can find an extended explanation addressing the aforementioned changes (lines 188201).

6) L136. I appreciate that the authors want to provide the details of the statistical analyses on female survival, but all the parentheses make this paragraph extremely difficult to follow. Perhaps including the results of the analysis as a table would make this much easier to read.

We have included a table to show the output from separate GLMs for each temperature level to explore significant interactions between temperature and mating system effects for lifespan (table 1).

7) L146. Again, this paragraph is complex and difficult to follow. The authors may want to state the main finding of their data, before detailing the statistics that support this finding. As for the analysis of the LRS, the P-values of the main effects are questionable if they have been adjusted for significant interaction (which is what is implied in the use of type III ANOVA, L560). As an alternative the authors could conduct posthoc tests on the effect of temperature or mating system for each mating system or temperature respectively, applying appropriate corrections for a multiplicity of tests, e.g. Bonferroni. As for Figure 1, the parameter values for the main effects in Figure 4 (right y-axis) would be best presented in a supplementary table.

For a better understanding, we have re-structured and reworded the Results section. While we utilized ANOVA type III to report the p-values. This approach cautions against direct interpretation of main effects, but we feel in this case interpretation is straightforward given the data trends depicted in the figures. However, we conducted further analyses by running Tukey's post hoc tests on the impact of the mating system at each temperature, as shown in table S1.1b for reproductive behaviour; and results are in accordance with our interpretation. We now present all the figures without the right y-axis and provide tables to show the size effects (tables 1-3)

8) I find the evidence that temperature affects ejaculate quality to harm females much less convincing. The authors report data on the effects of the mating system on mating duration and remating latency in females mating with males that have been kept singularly (low sperm competition) or in groups of three (high sperm competition). They should clearly explain the relationship between sperm competition, ejaculate quality, and these two assays, with citations if these methods have been used before. The description of these experiments is particularly difficult to follow, but it appears that they recorded mating duration both for the first mating (with males having just been exposed to different levels of perceived sperm competition) and during rematings. Which of these is shown in Figure 5A? They also conducted mating assays on males that had been maintained at different temperatures for 48h or 13 days (L 502). Data from which of these are used in Figure 5A? Which of these were used for female remating latency? Because the experimental methods are so difficult to follow, it precludes an interpretation of the data and makes it difficult to determine whether the data support the conclusions of the authors.

We are sorry that our initial structure was not clear enough to show the overall experimental design. We now provide more detailed explanations regarding the methods and results. We also re-made figure 5 (now figure 6), showing the data for mating duration for the first mating (Figure 6a) and female remating latency (Figure 6b) for the two treatment durations separately.

9) L179: This is another dense paragraph that essentially shows that sperm competition risk does not affect female fecundity and survival. That is, while sperm competition may increase mating duration, or remating latency, these do not appear to result in female harm. This undermines the interpretation that changes in mating duration and remating latency with risk of sperm competition reflect the chemistry of the ejaculate. An alternative explanation is that maintaining males with other males may increase his mating duration, which in turn leads to an increase in remating latency. It is interesting to note that the interaction between temperature and risk of sperm competition on mating duration and remating latency reflects a decline in these dependent factors with temperature when risk is low, but maintenance when risk is high. Chemical analysis of the ejaculate would help clarify the relationship between temperature and sperm competition on post-copulatory female harm.

We have attempted to streamline and clarify the female fecundity and survival part of the manuscript so hopefully the message we wanted to get across is clearer (lines 444-466). We have also clarified our results in relation to the ejaculate effects on female receptivity (lines 423-437). Briefly, sperm competition risk (i.e., keeping males in a group of males vs. isolation) does affect both mating duration and the effects of mating on female re-mating latency, but the key is that these effects appear to be higher at 28ºC than at other temperatures, which suggests modulation of both these variables at warmer temperatures, as pointed out by the reviewer. Clearly, the reviewer is correct in pointing out that these results are intriguing but cannot be interpreted conclusively without analysis of temperature effects on the composition of ejaculates. We discuss this along with potential explanations in the current version, where we have attempted to clarify what we did, why we did it and what this may mean, but erring on the side of caution (lines 633-655).